# Understanding Benefit of Personalization: Beyond Classification

## Abstract

In many applications spanning healthcare, finance, and admissions, it is beneficial to have personalized machine learning models that make predictions tailored to subgroups. This can be achieved by encoding personalized characteristics (such as age and sex) as model inputs. In domains where model trust and accuracy are paramount, it is critical to evaluate the effect of personalizing models not only on prediction accuracy but also on the quality of post-hoc model explanations. This paper introduces a unifying framework to quantify and validate personalization benefits in terms of both prediction accuracy and explanation quality across different groups, extending this concept to regression settings for the first time –broadening its scope and applicability. For both regression and classification, we derive novel bounds for the number of personalized attributes that can be used to reliably validate these gains. Additionally, through our theoretical analysis we demonstrate that improvements in prediction accuracy due to personalization do not necessarily translate to enhanced explainability, underpinning the importance to evaluate both metrics when applying machine learning models to safety-critical settings such as healthcare. Finally, we evaluate our proposed framework and validation techniques on a real-world dataset, exemplifying the analysis possibilities that they offer. This research contributes to ongoing efforts in understanding personalization benefits, offering a robust and versatile framework for practitioners to holistically evaluate their models.

## 1 Introduction

To prevent discrimination, protected attributes like sex, race, or religion are frequently restricted in sensitive decision-making processes, such as employment (U.S. Equal Employment Opportunity Commission, 1963), lending, education, and healthcare. These attributes are legally safeguarded, often due to a history of bias or unequal treatment. However, in some applications, taking these demographic factors into account can significantly improve prediction performance. This is especially true in medicine, where using protected attributes can enhance clinical prediction models by accounting for biological and sociocultural differences affecting health outcomes. For example, cardiovascular disease risk prediction models often improve when including sex (Paulus et al., 2016; Huang et al., 2024; Mosca et al., 2011) and race (Paulus et al., 2018), as men and women exhibit distinct heart disease risk patterns, and racial differences –such as increased hypertension prevalence in African Americans– are crucial for accurate risk assessment.

However, such sensitive attributes are known to increase bias in machine learning models (Kodiyan, 2019), so practitioners must ensure that they provide clear performance gains across all involved subgroups before adopting them. In fact, while incorporating sensitive data often increases overall accuracy, previous studies have already shown that *personalization* does not uniformly improve performance across all population subgroups (Suriyakumar et al., 2023). To rigorously measure personalization quality and fairness, the work of Monteiro Paes et al. (2022) introduced the *Benefit of Personalization* (BoP) metric to quantify personalization gain in terms of model classification prediction, based on comparing personalized model performance to that of a generic model trained without group attributes. Additionally, they derive a practical information-theoretic limit on error probability for classification tasks.

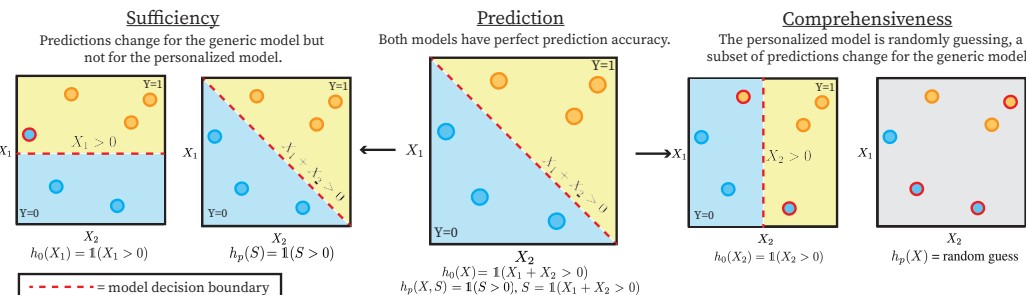

Figure 1: Practioners should not dismiss personalized models just because they do not provide a clear BoP gain in terms of prediction accuracy. We illustrate a toy example when, in such a context, there is a BoP gain in terms of explainability (concept introduced in this work). On a classification task, we compare a personalized model $h_p$ that uses group attributes with a generic model $h_0$ without them, both of them achieving perfect accuracy. Explanations for both models are generated, producing a subset of input features that contribute most to the model's predictions. For $h_0$ this subset is $X_1$, for $h_p$ this subset is $S$. We evaluate the quality of these explanations using the two widely-used criteria of sufficiency and comprehensiveness, which measure how original predictions change when only using or excluding the important features, respectively. We observe that $h_p$ produces a lower sufficiency and higher comprehensiveness than $h_0$, reflecting an improvement in explanation quality.

Nevertheless, BoP metric has not yet been extended or bounded for *regression* models, which in many inherently continuous processes can capture patterns that might be overlooked if data were classified into discrete categories. For example, in the medical domain, instead of classifying glucose levels as "low", "normal", or "high", regression can predict exact blood glucose levels in diabetes patients, enabling more accurate insulin dosing and management (Butt et al., 2023). In addition to this, no previous work has explored and audited personalization's effect on model *explainability*, a necessity for clinical decision-making and patient trust. For instance, in a study of pneumonia risk, machine learning models counter-intuitively predicted lower mortality risk for patients with asthma, and this finding actually reflected that asthma patients often received more aggressive care, lowering their risk (Caruana et al., 2015). Without explainability methods to understand how models make predictions, such insights could be missed, potentially leading to inadequate treatment decisions.

**Contributions.** This work addresses the previous points, aiming at gaining a wider and more comprehensive understanding of the impact of using sensitive characteristics in machine learning models. In particular, the main contributions of this paper include:

- We propose a generalized BoP framework to evaluate both model explanation quality and prediction accuracy across classification and regression settings. This extension not only broadens BoP scope and applicability, but also introduces a novel analysis on how protected attributes affects model's explainability (see Fig. 1). This approach is particularly valuable in contexts where understanding model decisions is as critical as the decisions themselves. (Section 4)

- We prove rigorous statistical bounds for auditing the generalized BoP metrics in classification and regression contexts. To the best of our knowledge, our work is the first to prove such bound to evaluate BoP for regression. Further, our analysis improves the bounds for classification in the previous work Monteiro Paes et al. (2022). (Section 5)

- Our theoretical analyses yield new insights into BoP across different settings. We demonstrate that regression models can potentially utilize more group attributes than classification models while keeping low testing error. Furthermore, we uncover a critical incompatibility: improvements in prediction accuracy from personalization do not necessarily correlate with enhanced explainability, underscoring the importance of evaluating both criteria in models where accuracy and interpretability are paramount. (Section 5)

- We apply our framework and validation tests on a real-world dataset for a classification and a regression task. In particular, our experimental results empirically demonstrate that personalization can indeed affect accuracy and explainability differently. (Section 6)

## 2 RELATED WORKS

**Personalization** Our research is part of a body of work that investigates how the use of personalized features in machine learning models influences group fairness outcomes (Suriyakumar et al., 2023). Monteiro Paes et al. (2022) defined a metric to measure the smallest gain in accuracy that any group can expect to receive from a personalized model. The authors demonstrate how this metric can be employed to compare personalized and generic models, identifying instances where personalized models produce unjustifiably inaccurate predictions for subgroups that have shared their personal data. However, this literature has focused on the classification framework and has not been generalized to regression tasks. Furthermore, this work has been solely concerned with evaluating how model accuracy is affected, and has not explored how personalizing a model affects the quality of its explanations.

**Explainability** Typical approaches to model explanation involve measuring how much each input feature contributes to the model's output, highlighting important inputs to promote user trust. This process often involves using gradients or hidden feature maps to estimate the importance of inputs (Simonyan et al., 2014; Smilkov et al., 2017; Sundararajan et al., 2017; Yuan et al., 2022). For instance, gradient-based methods use backpropagation to compute the gradient of the output with respect to inputs, with higher gradients indicating greater importance(Sundararajan et al., 2017; Yuan et al., 2022). The quality of these explanations is often evaluated using the principle of *faithfulness* (Lyu et al., 2024; Dasgupta et al., 2022; Jacovi & Goldberg, 2020), which measures how accurately an explanation represents the reasoning of the underlying model. Two key aspects of faithfulness are *sufficiency* and *comprehenesiveness* (DeYoung et al., 2020; Yin et al., 2022); the former assesses whether the inputs deemed important are adequate for the model's prediction, and the latter examines if these features capture the essence of the model's decision-making process.

**Personalization on Explainability** The field of the effects of personalization on explainable machine learning is largely unexplored. Previous work has investigated gaps in fidelity across subgroups and found that the quality and reliability of explanations may vary across different subgroups (Balagopalan et al., 2022). The work Balagopalan et al. (2022) trains a human-interpretable model to imitate the behavior of a blackbox model, and characterizes fidelity as how well it matches the blackbox model predictions. To achieve fairness parity, this paper explored using only features with zero mutual information with respect to a protected attribute. However, it left feature importance explanations out of its scope. Additionally, this work neither considers regression tasks nor looks at how personalization affects differences in explanation quality across subgroups.

We extend related works tackling fairness in regression in Appendix Section A.

## 3 BACKGROUND AND PROBLEM SETTING

This section reviews relevant concepts and methodologies in the fields of personalization and explainability, laying the groundwork to present and contextualize our contributions.

---

**Notation.** In what follows, let $\mathcal{X}, \mathcal{S}, \mathcal{Y}$ denote, respectively, the feature, group attributes and label spaces. Additionally, we denote an auditing dataset by

$$\mathcal{D} = \{(\mathbf{x_i}, \mathbf{s_i}, y_i)\}_{i=1}^{N},$$

where $N$ is the total number of samples and, for each sample $i$, $\mathbf{x_i} \in \mathcal{X}$ represents its feature vector, $\mathbf{s_i} \in \mathcal{S}$ its vector of group attributes, and $y_i \in Y$ the corresponding label or target.

---

**Supervised learning and personalization.** Within a supervised learning setting, a personalized model $h_p : \mathcal{X} \times \mathcal{S} \to \mathcal{Y}$ aims to predict an outcome variable $Y \in \mathcal{Y}$ using both an input feature vector $X \in \mathcal{X}$ and a vector of group attributes $S \in \mathcal{S}$. In such a setting, we are interested in analyzing the benefits of personalization by comparing $h_p$ with a generic model $h_0 : \mathcal{X} \to \mathcal{Y}$ that does not use (sensitive) group attributes. We assume that these models are trained on a training dataset that is independent of the auditing dataset $\mathcal{D}$. The following definition enables us to measure the overall performance of the model with respect to a cost function, thus facilitating this comparison:

**Definition 1** (Cost). The cost of a model $h$ with respect to a cost function $\text{cost} : \mathcal{Y} \times \mathcal{Y} \to \mathbb{R}$ is defined as:

$$C(h) \triangleq \begin{cases} \mathbb{E}[\text{cost}(h(\mathbf{X}), Y)] & \text{if} \quad h : \mathcal{X} \to \mathcal{Y} \quad \text{(generic model)} \\ \mathbb{E}[\text{cost}(h(\mathbf{X}, \mathbf{S}), Y)] & \text{if} \quad h : \mathcal{X} \times \mathcal{S} \to \mathcal{Y} \quad \text{(personalized model)} \end{cases} \tag{1}$$

Analogously, $\hat{C}$ is an empirical estimate of $C$, e.g., $\hat{C}(h_0) = \frac{1}{N} \sum_{i=1}^{N} \text{cost}(h(\mathbf{x}_i), y_i)$.[1]

Since we are defining a framework that seeks to minimize cost, any chosen cost function should satisfy the principle of "lower cost means better performance". Moreover, we note that this definition can be easily extended and applied to different groups:

**Definition 2** (Group Cost). The group cost, of a model $h$ for group $\mathbf{s} \in \mathcal{S}$ with respect to a cost function $\text{cost} : \mathcal{Y} \times \mathcal{Y} \to \mathbb{R}$ is defined as:

$$C_s(h, \mathbf{s}) \triangleq \begin{cases} \mathbb{E}[\text{cost}(h(\mathbf{X}), Y) \mid \mathbf{S} = \mathbf{s}] & \text{if} \quad h : \mathcal{X} \to \mathcal{Y} \quad \text{(generic model)} \\ \mathbb{E}[\text{cost}(h(\mathbf{X}, \mathbf{s}), Y) \mid \mathbf{S} = \mathbf{s}] & \text{if} \quad h : \mathcal{X} \times \mathcal{S} \to \mathcal{Y} \quad \text{(personalized model)} \end{cases} \tag{2}$$

**Evaluating Predictions.** Previous cost definitions can be applied to evaluate model performance on a prediction task. In this case, the cost function can be either the loss function $\ell_{\text{train}}$ used for training, or an auxiliary evaluation of performance $\ell_{\text{eval}}$. Most of the time, we will not distinguish between $\ell_{\text{train}}$ and $\ell_{\text{eval}}$ and refer to this function as the loss $\ell$, such that we have:

$$\text{cost}(h, \mathbf{x}, y) \triangleq \begin{cases} \ell(h(\mathbf{x}), y) & \text{if} \quad h : \mathcal{X} \to \mathcal{Y} \quad \text{(generic model)} \\ \ell(h(\mathbf{x}, \mathbf{s}), y) & \text{if} \quad h : \mathcal{X} \times \mathcal{S} \to \mathcal{Y} \quad \text{(personalized model)}, \end{cases} \tag{3}$$

where:

$$\ell(y, \hat{y}) \triangleq \begin{cases} \|y - \hat{y}\|^2 & \text{if squared error loss (regression)} \\ \mathbb{1}(y \neq \hat{y}) & \text{if 0-1 loss (binary classification)} \\ \text{other loss functions} & \text{if alternative models or custom losses.} \end{cases} \tag{4}$$

By plugging $\ell$ into Def. 1 and 2, one can empirically evaluate model prediction performance across all samples, and across subsets of samples designated by shared group attributes. This can be done for generic ($h_0$) or personalized models ($h_p$).

**Evaluating Explainability.** Notably, previous cost definitions can also be applied to evaluate the explainability of a model. Here, we focus on the subset of explainability techniques that output an importance score for each model input –this importance score quantifying the sensitivity between model inputs with regards to the model predictions.[2] In particular, to evaluate explanations with our framework, we use the cost functions of either comprehensiveness or sufficiency of a model's explanation (DeYoung et al., 2020), generalizing them to use any loss between predicted values. These functions measure the quality of an explanation based on removing or only keeping important features:[3]

---

**Notation.** An explanation $E$ for a model $h$ is defined as:

$$E = \begin{bmatrix} \mathbf{e_1} & \mathbf{e_2} & \dots & \mathbf{e_N} \end{bmatrix}, \tag{5}$$

where each vector $\mathbf{e_i}$ denotes the importance for each input feature of the $i$-th sample. For each sample $i$, we find the top $r$ features with the highest importance, so that we get $J_i = \{j_1, \cdots, j_r\}$, representing the indices of the $r$ largest values in $\mathbf{e_i}$.

- For a generic model $h_0$, we denote by $\mathbf{X}_{\setminus J}$ the feature input when removing the top $r$ most important features, and by $\mathbf{X}_J$ the complement –only keeping the $r$ most important ones.

- Analogously, for a personalized model $h_p$, the top $r$ most important features across $\mathbf{X} \cup \mathbf{S}$ are either removed or selected. We denote the the resulting features+attributes set with top features removed by $\mathbf{S}_{\setminus J}$, and with only the top features kept by $\mathbf{S}_J$.

---

[1]All other empirical definitions, including for individual samples, can be found in Appendix B and C.

[2]However, the way importance scores are found differs per explanation method –please refer to Yuan et al. (2022) for a review of possible options.

[3]Removing/disregarding a feature simply means setting it to 0 (Ancona et al., 2018).

**Definition 3** (Incomprehensiveness). Incomprehensiveness measures the change in model prediction when removing the top features[4]:

$$\text{cost}(h, \mathbf{x}) \triangleq \begin{cases} -\ell(h(\mathbf{x}), h(\mathbf{x}_{\setminus J})) & \text{if} \quad h : \mathcal{X} \to \mathcal{Y} \quad \text{(generic model)} \\ -\ell(h(\mathbf{x}, \mathbf{s}), h(\mathbf{x}_{\setminus J}, \mathbf{s}_{\setminus J})) & \text{if} \quad h : \mathcal{X} \times \mathcal{S} \to \mathcal{Y} \quad \text{(personalized model)} \end{cases} \tag{6}$$

where $\ell$ is a measure of prediction performance as defined in Equation 4. A large negative incomprehensiveness score is desired, as it shows that removing the inputs most relevant to the explanation significantly alters the prediction.

Sufficiency can be defined in a similar manner, but rather than removing the top $r$ features, only the top $r$ features are preserved.

**Definition 4** (Sufficiency). In the case of sufficiency, the cost function can be defined as follows:

$$\text{cost}(h, x) \triangleq \begin{cases} \ell(h(x), h(\mathbf{x}_J)) & \text{if} \quad h : \mathcal{X} \to \mathcal{Y} \quad \text{(generic model)} \\ \ell(h(\mathbf{x}, \mathbf{s}), h(\mathbf{x}_J, \mathbf{s}_J)) & \text{if} \quad h : \mathcal{X} \times \mathcal{S} \to \mathcal{Y} \quad \text{(personalized model)} \end{cases} \tag{7}$$

where $\ell$ is a measure of prediction performance as defined in Equation 4.[5] A low sufficiency score is desired to verify that the inputs deemed important are sufficient for the prediction.

# 4 A Generalized Framework for Benefit of Personalization

Leveraging the cost definitions from Section 3, this section introduces a novel generalized approach to quantify the Benefit of Personalization (BoP) –i.e. to rigorously measure whether a personalized model ($h_p$) performs better than its generic counterpart ($h_0$). Drawing inspiration from Monteiro Paes et al. (2022), we propose the first BoP framework that *(i)* incorporates explainability into the analysis (apart from prediction accuracy), and that *(ii)* spans both regression and classification tasks.

We start by defining some relevant BoP concepts and metrics.

> **Notation.** In what follows, we consider that a fixed data distribution $P_{\mathbf{X}, \mathbf{S}, Y}$ is given, and that $h_0$ and $h_p$ models minimize the loss over the training dataset $\mathcal{D}_{train}$.

**Definition 5** (Population BoP). The gain from personalizing a model can be measured by comparing the costs of the generic and personalized models:

$$\text{BoP}(h_0, h_p) \triangleq C(h_0) - C(h_p). \tag{8}$$

**Definition 6** (Groupwise BoP). Similarly, the gain from personalizing a model across each subgroup of samples can be obtained by:

$$\text{BoP}_s(h_0, h_p, \mathbf{s}) \triangleq C_s(h_0, \mathbf{s}) - C_s(h_p, \mathbf{s}). \tag{9}$$

Therefore, Groupwise BoP can be measured across all sensitive subgroups to understand exactly how personalization affects each one of them. In fact, it is crucial to consider if personalization benefits each subgroup equally, and more so to investigate whether personalization actively harms particular subgroups (Monteiro Paes et al., 2022). The following concept is useful to identify the latter scenario:

**Definition 7** (Minimal Group BoP).

$$\gamma(h_0, h_p) \triangleq \min_{\mathbf{s} \in \mathcal{S}} (\text{BoP}_s(h_0, h_p, \mathbf{s})) \tag{10}$$

In particular, note that a positive Minimal Group BoP indicates that all subgroups receive better performance with respect to the cost function. Contrary to this, a negative value reflects that at least one group is disadvantaged by the use of personal attributes. When the Minimal Group BoP is small or negative, the practitioner should reconsider the use of personalized attributes in terms of the trustworthiness of the model for all subgroups.

In the following subsections we show how these abstract definitions can be used to measure BoP for both predictions and explanations, each across both classification and regression tasks.

---

[4]Note that we negate the traditional notion of comprehensiveness, and propose the metric of incomprehensiveness, because we define our cost metrics such that lower cost means better performance.

[5]Note that in these definitions our focus is in explaining the model rather than the phenomenon Amara et al. (2024). These definitions can be written for explanation of phenomena by replacing $h(\mathbf{x})$ for the generic model and $h(\mathbf{x}, \mathbf{s})$ for the personalized model with $y$ in Equations 6 and 7.

### 4.1 BoP for Prediction (BoP-P)

When analyzing BoP in terms of prediction accuracy, the main concern is to analyze how performance differs across subgroups. We show how the Minimal Group BoP can be expressed for classification and regression tasks (given a particular choice of loss function in each case).

**Classification** In the binary classification case, using the 0-1 loss function $\ell(y, h(\mathbf{x}, \mathbf{s})) \triangleq \mathbf{1}[y \neq h(\mathbf{x}, \mathbf{s})]$, the Minimal Group BoP is:

$$\gamma_{BOP-P}(h_0, h_p; \mathcal{D}) = \min_{\mathbf{s} \in \mathcal{S}} \left( \Pr\left(h_0(\mathbf{X}) \neq Y \mid \mathbf{S} = \mathbf{s}\right) - \Pr\left(h_p(\mathbf{X}, \mathbf{s}) \neq Y \mid \mathbf{S} = \mathbf{s}\right) \right) \in [-1, 1].$$

In this setting, the Minimal Group BoP measures the minimum gain in accuracy between $h_p$ and $h_0$.

**Regression** In the regression case, using the square error loss function, the Minimal Group BoP is:

$$\gamma_{BOP-P}(h_0, h_p; \mathcal{D}) = \min_{\mathbf{s} \in \mathcal{S}} \left( \mathbb{E}\left[\|h_0(\mathbf{X}) - Y\|^2 \mid \mathbf{S} = \mathbf{s}\right] - \mathbb{E}\left[\|h_p(\mathbf{X}, \mathbf{s}) - Y\|^2 \mid \mathbf{S} = \mathbf{s}\right] \right)$$

$$\in [-\infty, +\infty].$$

### 4.2 BoP for Explainability (BoP-X)

Lastly, we introduce novel and practical definitions of BoP for explainability, leveraging the incomprehensiveness and sufficiency cost functions. It is recommended that practioners apply both metrics to understand the effects of personalization in terms of faithfulness as a whole. For the sake of space, we only show expressions of the Minimal Group BoP in terms of sufficiency—both for classification and regression—but the analogous incomprehensiveness expressions can be found in Appendix D.

**Classification** In the classification case, with the 0-1 loss function and using the cost function defined for sufficiency, the Minimal Group BoP can be written as:

$$\gamma_{BOP-X}(h_0, h_p; \mathcal{D}) = \min_{\mathbf{s} \in \mathcal{S}} \left( \Pr\left(h_0(\mathbf{X}) \neq h_0(\mathbf{X}_J) \mid \mathbf{S} = \mathbf{s}\right) \right.$$

$$\left. - \Pr\left(h_p(\mathbf{X}, \mathbf{s}) \neq h_p(\mathbf{X}_J, \mathbf{s}) \mid \mathbf{S} = \mathbf{s}\right) \right), \quad \text{where} \quad \gamma \in [-1, 1].$$

**Regression** Using the cost function defined for sufficiency with the square error loss function, the Minimal Group BoP in the case of regression can be written as:

$$\gamma_{BOP-X}(h_0, h_p; \mathcal{D}) = \min_{\mathbf{s} \in \mathcal{S}} \left( \mathbb{E}\left[\|h_0(\mathbf{X}) - h_0(\mathbf{X}_J)\|^2 \mid \mathbf{S} = \mathbf{s}\right] \right.$$

$$\left. - \mathbb{E}\left[\|h_p(\mathbf{X}, \mathbf{s}) - h_p(\mathbf{X}_J, \mathbf{s}_J)\|^2 \mid \mathbf{S} = \mathbf{s}\right] \right), \quad \text{where} \quad \gamma \in [-\infty, +\infty].$$

## 5 Statistical Tests for Generalized BoP

Calculating the BoP requires exact knowledge of the data distribution, a condition rarely met in practice. Moreover, within the ubiquitous finite sample regime, it is critical to understand the feasibility of the empirical BoP –given for instance a limited sample size, or a large number of group attributes. In this section, drawing inspiration from Monteiro Paes et al. (2022), we first introduce a hypothesis testing framework to assess whether a personalized model yields a substantial performance improvement across all groups. Subsequently, we derive a novel information-theoretic bound on the reliability of this procedure, both for binary and real-valued cost functions. In addition to this, we investigate how the different BoP metrics of our framework relate to each other (classification vs. regression, prediction vs. explainability), which leads to new insights into BoP.

> All proofs for subsequent theorems, lemmas and corollaries can be found in Appendix Sections E.1, E.2,E.3,F and H.

**Hypothesis Test** Given a personalized classifier $h_p$, a generic classifier $h_0$, and auditing dataset $\mathcal{D}$, we verify whether using a personalized model $h_p$ yields an $\epsilon > 0$ gain in expected performances compared to using the generic model $h_0$. Note that the improvement $\epsilon$ is in cost function units, and corresponds to the reduction in cost for the group for which moving from $h_0$ to $h_p$ is least

advantageous –i.e. $\epsilon$ actually represents the improvement for the group that benefits the least from the personalized model. In this context, we propose the following hypothesis test:

$$H_0: \quad \gamma(h_0, h_p; \mathcal{D}) \leq 0 \quad \Leftrightarrow \quad \text{Personalized } h_p \text{ does not bring any gain for at least one group,}$$

$$H_1: \quad \gamma(h_0, h_p; \mathcal{D}) \geq \epsilon \quad \Leftrightarrow \quad \text{Personalized } h_p \text{ yields at least } \epsilon \text{ improvement for all groups.}$$

To actually perform this hypothesis test, we follow (Monteiro Paes et al., 2022) and propose the following threshold test on the estimate of the BoP (i.e., the empirical BoP $\hat{\gamma}$):

$$\hat{\gamma} \geq \epsilon \Rightarrow \text{Reject } H_0: \text{ Conclude that personalization yields at least } \epsilon \text{ improvement for all groups.}$$

Furthermore, we characterize the reliability of hypothesis tests in terms of their probability of error. We define the probability of error $P_e$ of the hypothesis test on $H_1$ and $H_0$ as:

$$P_e = \Pr(\text{Type I error}) + \Pr(\text{Type II error})$$
$$= \Pr(\text{Rejecting } H_0 | H_0 \text{ is true}) + \Pr(\text{Failing to reject } H_0 | H_1 \text{ is true})$$

If this probability exceeds 50%, the test is no more reliable than the flip of a fair coin, making it too unreliable to support any meaningful verification. Therefore, it would be practical to compute a lower bound on the worst case scenario for this probability of error, so that if this lower bounds exceeds 50%, we would not trust the test. We precisely derive such bounds for binary and regression cost functions in the following paragraphs (applicable for both prediction and explainability).

---

**Notation.** We formalize our hypothesis test by an abstract *decision* function $\Psi : (h_0, h_p, \mathcal{D}, \epsilon) \rightarrow \{0, 1\}$ such that $\Psi(h_0, h_p, \mathcal{D}, \epsilon) = 1 \Rightarrow \text{Reject } H_0$.

---

**Testing the BoP: Binary Cost Function** The case of BoP for prediction in classification has been studied in (Monteiro Paes et al., 2022). As the authors noted, the theorems and proofs can be generalized to any scenario where the individual cost can be described by a Bernouilli random variable –i.e., where the cost function takes values in $\{0, 1\}$, and consequently the individual BoP can be described by a categorical random variable with values in $\{-1, 0, 1\}$. The next theorem refines Theorem 1 of (Monteiro Paes et al., 2022) to provide a tighter lower bound:

**Theorem 1** (Lower bound for categorical individual BoP). *The lower bound writes:*

$$\min_{\Psi} \max_{\substack{P_{\mathbf{X}, \mathbf{S}, Y} \in H_0 \\ Q_{\mathbf{X}, \mathbf{S}, Y} \in H_1}} P_e \geq 1 - \frac{1}{2\sqrt{d}} \left(1 + 4\epsilon^2\right)^{m/2} \tag{11}$$

*where $P_{\mathbf{X}, \mathbf{S}, Y}$ is a distribution of data, for which the generic model $h_0$ performs better, i.e., the true $\gamma$ is such that $\gamma(h_0, h_p, \mathcal{D}) < 0$, and $Q_{\mathbf{X}, \mathbf{S}, Y}$ is a distribution of data points for which the personalized model performs better, i.e., the true $\gamma$ is such that $\gamma(h_0, h_p, \mathcal{D}) \geq \epsilon$. Dataset $\mathcal{D}$ is drawn from an unknown distribution and has $d$ groups where $d = 2^k$, with each group having $m = \lfloor N/d \rfloor$ samples.*

**Testing the BoP: Real-valued Cost Function** Focusing next on regression tasks, we generalize the previous discrete-domain theory to continuous cost functions. In particular, we derive from scratch new lower bounds to any scenario where the individual BoP can be described by a Normal random variable.[6] Assuming that the value of $\epsilon$ is fixed, we provide the following theorem:

**Theorem 2** (Lower bound for Gaussian individual BoP). *The lower bound writes:*

$$\min_{\Psi} \max_{\substack{P_{\mathbf{X}, \mathbf{S}, Y} \in H_0 \\ Q_{\mathbf{X}, \mathbf{S}, Y} \in H_1}} P_e \geq 1 - \frac{1}{2\sqrt{d}} \exp\left(\frac{\epsilon^2}{\sigma^2}\right)^{m/2}$$

*where $P_{\mathbf{X}, \mathbf{S}, Y}$ is a distribution of data, for which the generic model $h_0$ performs better, i.e., the true $\gamma$ is such that $\gamma(h_0, h_p, \mathcal{D}) < 0$, and $Q_{\mathbf{X}, \mathbf{S}, Y}$ is a distribution of data points for which the personalized model performs better, i.e., the true $\gamma$ is such that $\gamma(h_0, h_p, \mathcal{D}) \geq \epsilon$. Dataset $\mathcal{D}$ is drawn from an unknown distribution and has $d$ groups, with each group having $m = \lfloor N/d \rfloor$ samples. $\sigma$ is the standard deviation of the BoP across participants, and is assumed to be the same across all groups.*

By leveraging the lower bounds provided by Theorems 1 and 2, the remainder of this section aims to answer how the different settings of our BoP framework connect and relate to each other.

---

[6]We additionally derive the bounds assuming the individual BoP can be described by a Laplacian distribution. The corresponding theorems and proof are provided in Appendix Section E.3.

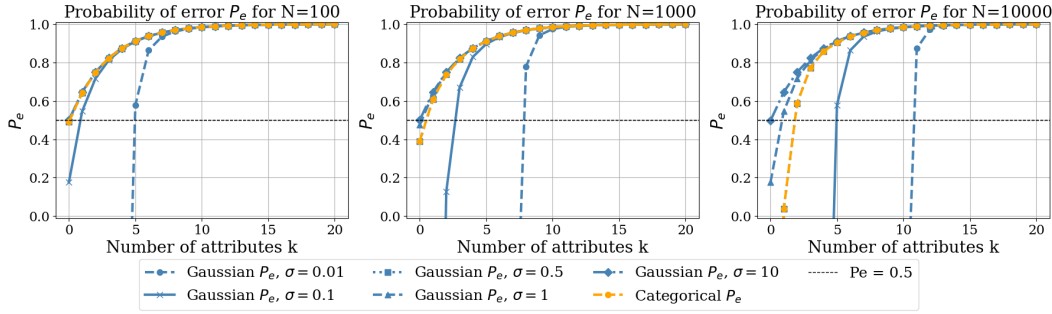

Figure 2: Probability of error $P_e$ versus number of attributes $k$ defining the number of groups $d = 2^k$ for varying number of samples $N$. In orange you see the line for a binary cost function and in blue you see the values real-valued cost function for varying values of $\sigma$. In all cases, $\epsilon = 0.01$. When $\sigma = 0.5$, the exponential term in the lower bound for real-valued $P_e$ becomes $4\epsilon^2$, which can be approximated as $1 + 4\epsilon$ for small $\epsilon$. Hence, we see the categorical BoP aligns with the real valued Pe for $\sigma = 0.5$. We see that for small $\sigma$ values in the real-valued case, the number of attributes $k$ that can be used before surpassing $P_e \geq 1/2$ is higher than for the categorical case.

**Does the maximum number of sensitive attributes allowed differ in Classification versus Regression?** Given the obtained bounds, we can compute the maximum number of attributes $k$ for which such a hypothesis test would make sense. To this end, we first prove that the lower bounds are a increasing function of $k$:

**Lemma 1.** *Given values of $\epsilon, n, \sigma$ fixed, the lower bounds in Theorems 1- 2 are monotonically increasing functions of $k$, the number of sensitive attributes defining the number of groups $d = 2^k$.*

This result was known for the binary case, but we also prove it for the real-valued case. The following results easily follow from the lemma:

**Corollary 1** (Maximum number of attributes (binary cost function))**.** *If we wish to maintain a probability of error such that $\min \max P_e \leq 1/2$, then the number of attributes $k$ should be chosen below a value $k_{\max}$ that depends on the number of samples $N$:*

$$k_{max} \leq 1.4427 W(N \log(4\epsilon^2 + 1)), \tag{12}$$

*where $W$ is the Lambert W function.*

**Corollary 2** (Maximum number of attributes (real-valued cost function))**.** *If we wish to maintain a probability of error such that $\min \max P_e \leq 1/2$ then the number of attributes $k$ should be chosen below a value $k_{\max}$ that depends on the number of samples $N$ and on the value of $\sigma$.*

$$k_{max} \leq 1.4427 W(\frac{\epsilon^2 N}{\sigma^2}) \tag{13}$$

*where $W$ is the Lambert W function.*

**Corollary 3** (Maximum attributes (real-valued cost function) for all people)**.** *See Appendix I.*

To better contextualize these theoretical results, Figure 2 plots the relation between $k$ and $P_e$ for a binary and a real-valued cost function, considering common sample sizes in medical applications. Looking at the number of attributes $k$ allowed for $P_e < 0.5$, we clearly observe the consequences of the extra dependence on $\sigma$ in the real-valued case. In particular, note that it allows a higher number of attributes than the binary case for small $\sigma$ values. This tells a much more subtle story than for the classification case, because the maximum number of attributes allowed depends on the distribution of the BoP across participants.

**Does the maximum number of sensitive attributes allowed differ in Prediction versus Explainability?** Within the setting of classification, the maximum number of sensitive attributes allowed does not differ in prediction versus explainability given that both utilize an individual cost function that can be described by a Bernoulli random variable. Within the setting of regression, the maximum

attributes can differ between prediction and explainability (for both sufficiency and incomprehensiveness) because it is dependent on the standard deviation of the BoP across participants. Therefore, the number of allowed attributes will differ provided this value is different for each criteria evaluated.

**Does BoP for Prediction Imply BoP for Explainability and Vice-Versa?** Finally, we examine the relationship between BoP-P and BoP-X. In the following theorem, we show that *the absence of BoP in terms of predictive accuracy does not necessarily imply the absence of benefits in terms of explainability.*

**Theorem 3.** *There exists $P_{\mathbf{X},\mathbf{S},Y}$ such that BoP-P$(h_0, h_p) = 0$ and BoP-X$(h_0, h_p) > 0$*

This theorem emphasizes the necessity of assessing the BoP in terms of both predictive accuracy (BoP-P) and explainability (BoP-X). A personalized model may not demonstrate superior predictive performance yet still improve explainability. Evaluating personalized models solely on predictive accuracy risks can overlook substantial gains in interpretability—see Figure 1 for a visual example.

For a simple additive model, we can show that *BoP-X $= 0$ does imply BoP-P $= 0$*. Note that, by BoP-X $= 0$, we mean both sufficiency and comprehensiveness do not improve with personalization. Proving this for a general class of model remains an open question.

**Lemma 2.** *Assume that $h_0$ and $h_p$ are Bayes optimal classifiers and $P_{\mathbf{X},\mathbf{S},Y}$ follows an additive model, i.e.,*

$$Y = \alpha_1 X_1 + \cdots \alpha_t X_t + \alpha_{t+1} S_1 + \cdots + \alpha_{t+k} S_k + \epsilon, \tag{14}$$

*where $X_1, \cdots, X_t$ and $S_1, \cdots, S_k$ are independent, and $\epsilon$ is an independent random noise. Then, if BoP-X$(h_0, h_p) = 0$, BoP-P$(h_0, h_p) = 0$.*

# 6 APPLYING THE FRAMEWORK

This section empirically evaluates the generalized framework introduced in Section 4 to classification and regression tasks. Additionally, we leverage the validation tools developed in previous Section 5 to analyze the reliability of our results (shown in Table 1).

**Datasets.** We apply our framework to the High School Longitudinal Study (HSLS) dataset (Rogers et al., 2018) utilizing two group attributes: $\text{Sex} \times \text{Race} \in \{\text{Female}, \text{Male}\} \times \{\text{White}, \text{NonWhite}\}$. We downsample the most prevalent groups so that all groups have roughly the same number of samples. For the regression task the goal is to predict the math IRT-estimated scale score. For the binary classification task, we predict if the student's score falls in the top 50% or bottom 50%. For both classification and regression, we fit two neural network models: one with a one-hot encoding of the group attributes ($h_p$), and the other without group attributes ($h_0$). Moreover, regression prediction values are normalized to have mean 0 and standard deviation 1.

**Explainability Method.** To generate model explanations, we use the Captum Integrated gradients explainer method (Sundararajan et al., 2017). This method calculates the gradient of the output with respect to the input for each subject, and scales the result to get a contribution value for each input feature. To evaluate BoP-X using sufficiency and incomprehensivess, we select an value $r$ such that 50% of features are kept or removed. Plots in Appendix J depict how sufficiency and incomprehensiveness change for different values of $r$, as well as show the individual BoP distribution.

**Experimental Results.** Table 1 shows full results of the Population, Groupwise, and Minimal Group BoP on the test set; the corresponding tables for the training dataset can be found in Appendix G. The 0-1 and square loss cost functions are used for classification and regression, respectively.

**Statistical Validation.** To better understand the reliability of our empirical results, in Figure 3 we visually compute the information-theoretic lower bounds on probability of error that our validation framework provides for this dataset and these tasks. In particular, we get that we can trust our results ($P_e > 0.5$) for (i) $\gamma_{BoP} > 0.035$ for all metrics in the classification task, and in the case of regression (ii) $\gamma_{BoP-P} > 0.02$ for prediction accuracy, (iii) $\gamma_{BoP-X} > 0.25$ for incomprehensiveness, and (iv) $\gamma_{BoP-X} > 0.19$ for sufficiency.

**BoP-P Analysis.** In the case of prediction accuracy, we observe that the personalized model $h_p$ assigns less accurate predictions to specific subgroups for both classification and regression tasks

| Group | $n$ | Classification | | | Regression | | |
|---|---|---|---|---|---|---|---|
| | | Prediction | Incomprehensiveness | Sufficiency | Prediction | Incomprehensiveness | Sufficiency |
| Female, NonWhite | 274 | 0.011 | -0.248 | -0.259 | 0.005 | 1.97 | 3.72 |
| Female, White | 287 | -0.063 | -0.272 | -0.254 | -0.005 | 1.72 | 3.60 |
| Male, NonWhite | 274 | 0.004 | -0.124 | -0.153 | 0.004 | 1.48 | 4.15 |
| Male, White | 301 | -0.070 | -0.199 | -0.189 | 0.014 | 1.56 | 3.37 |
| All Population | 1136 | -0.031 | -0.211 | -0.214 | 0.005 | 1.68 | 3.70 |
| **Minimal Group BoP** | 1136 | -0.070 | -0.272 | -0.259 | -0.005 | 1.48 | 3.37 |

Table 1: Experimental results on the test set of the considered dataset, for both classification and regression. All columns show the value of $\hat{C}(h_0) - \hat{C}(h_p)$ evaluated for the corresponding metric. Values that are worsened by $h_p$ are colored red.

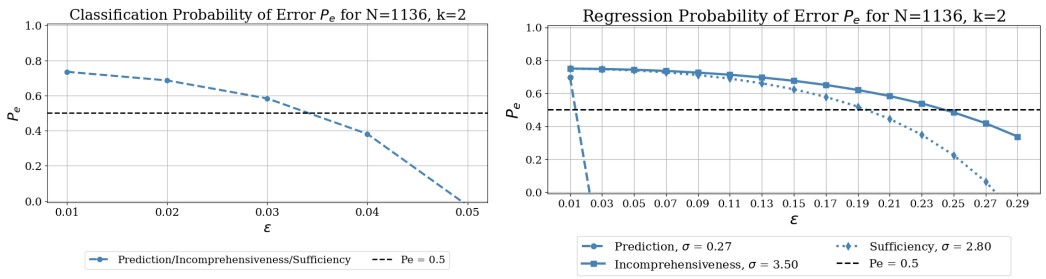

Figure 3: Leveraging the validation framework, we plot how the $P_e$ changes for different $\epsilon$ values for a set $N$ and $k$. On the left we use Theorem 1 for classification. On the right, Theorem 2 for regression (which has an additional dependency on $\sigma$, hence producing diferent results for each metric).

–even decreasing the overall accuracy for the entire population. Notably, the minimal BoP-P in classification exceeds 0.035, so we can conclude that in this case the use of sensitive attributes worsens accuracy. However, results are inconclusive for regression according to our statistical test.

**BoP-X Analysis.** In the case of explainability, we observe a clear difference in terms of the type of task. For classification, the personalized model worsens incomprehensiveness and sufficiency for all subgroups. In contrast, in the regression setting it increases sufficiency and incomprehensiveness across all of them. Additionally, in all explainability scenarios the statistical test is satisfied, so we can trust these observations.

**BoP-P vs. BoP-X.** For regression, the fact that the Minimal BoP for prediction is below 0.02 impedes us to draw conclusions about the BoP-P vs. BoP-X comparison. But, on the other hand, in classification we can conclude that sensitive attributes do worsen both explainability and prediction accuracy. However, despite the limited insights of these particular results, these experimental results exemplifies how this framework can be easily used to investigate the potential trade-offs between prediction and explainability in personalized models.

# 7 CONCLUDING REMARKS

This work introduces a novel BoP framework that accommodates model accuracy and explainability, both of which are paramount to building trust and transparency in sensitive settings. Additionally, the framework also extends the BoP analysis to regression tasks, enabling its application to new non-discretized scenarios. Through our theoretical analysis, we identified conditions for regression and classification where testing and estimation methods lack sufficient reliability to guarantee improvements across subgroups. Our findings also reveal that regression tasks have the potential to benefit from more personalized attributes than classification tasks, and that improved accuracy from personalization does not necessarily translate to enhanced explainability. Finally, and as exemplified by our evaluation, our framework and accompanying tests facilitate nuanced decisions regarding the use of protected attributes. Overall, this paper broadens the scope and applicability of BoP analysis and in doing so contributes to the selection of more fair and interpretable models.

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
