## A  EXTENDED RELATED WORKS: FAIRNESS IN REGRESSION

Our results are part of a growing body of work evaluating model fairness (Mehrabi et al., 2022). In the field of machine learning, the concept of fairness aims to mitigate biased outcomes affecting individuals or groups. Past works have defined individual fairness, which requires similar performance for similar individuals (Dwork et al., 2011), or group fairness (Dwork & Ilvento, 2019; Hardt et al., 2016) which seeks similar performance across different groups. Within machine learning fairness literature, the majority of methods, metrics, and analyses are predominantly intended for classification tasks, where labels take values from a finite set of values (Pessach & Shmueli, 2022). Among fair regression literature, multiple authors focus on designing fair learning methods rather than developing metrics for measuring fairness in existing models (Berk et al., 2017; Fukuchi et al., 2013; Pérez-Suay et al., 2017; Calders et al., 2013). Complimentary contributions focus on defining fairness criteria and establishing methods to evaluate fairness for regression tasks (Gursoy & Kakadiaris, 2022; Agarwal et al., 2019).

## B  EMPIRICAL DEFINITIONS

### B.1  COST

**Definition 8** (Group Cost). The empirical group cost, $\hat{C}_s(h, \mathbf{s})$, is defined as:

$$\hat{C}_s(h, \mathbf{s}) \triangleq \begin{cases} \frac{1}{n_\mathbf{s}} \sum_{i:\mathbf{s}_i=\mathbf{s}} \text{cost}\left(h\left(\mathbf{x}_i\right), y_i\right) & \text{if} \quad h : \mathcal{X} \to \mathcal{Y} \quad \text{(generic model)} \\ \frac{1}{n_\mathbf{s}} \sum_{i:\mathbf{s}_i=\mathbf{s}} \text{cost}\left(h\left(\mathbf{x}_i, \mathbf{s}_i\right), y_i\right) & \text{if} \quad h : \mathcal{X} \times \mathcal{S} \to \mathcal{Y} \quad \text{(personalized model)} \end{cases} \tag{15}$$

where $n_\mathbf{s}$ refers to the number of samples in group $\mathbf{s}$.

**Definition 9** (Individual Cost). The empirical individual cost, of a model $h$ for subject $i$ with respect to a cost function, $\text{cost} : \mathcal{Y} \times \mathcal{Y} \to \mathbb{R}$ is defined as:

$$\hat{C}_i(h, \mathbf{s}_i) \triangleq \begin{cases} \text{cost}(h(\mathbf{x}_i), y_i) & \text{if} \quad h : \mathcal{X} \to \mathcal{Y} \quad \text{(generic model)} \\ \text{cost}(h(\mathbf{x}_i, \mathbf{s}_i), y_i) & \text{if} \quad h : \mathcal{X} \times \mathcal{S} \to \mathcal{Y} \quad \text{(personalized model)} \end{cases} \tag{16}$$

## C  BOP

**Definition 10** (BoP). The empirical BoP is defined as:

$$\hat{\text{BoP}}(h_0, h_p) \triangleq \hat{C}(h_0, \mathbf{X}, Y) - \hat{C}(h_p, \mathbf{X}, \mathbf{S}, Y). \tag{17}$$

**Definition 11** (Group BoP). The empirical group BoP is defined as:

$$\hat{\text{BoP}}_s(h_0, h_p, \mathbf{s}) \triangleq \hat{C}_s(h_0, \mathbf{X}, Y) - \hat{C}_s(h_p, \mathbf{X}, \mathbf{s}, Y). \tag{18}$$

**Definition 12** (Minimal Group BoP). Empirical Minimal Group BoP

$$\hat{\gamma}(h_0, h_p; \mathcal{D}) \triangleq \min_{\mathbf{s} \in \mathcal{S}} (\hat{\text{BoP}}_s(h_0, h_p, \mathbf{s})) \tag{19}$$

**Definition 13** (Individual BoP). The gain any individual sample benefits from using personalized attributes is empirically written as:

$$\hat{\text{BoP}}_i(h_0, h_p) = \hat{C}_i(h_0, x_i, y_i) - \hat{C}_i(h_p, x_i, s_i, y_i). \tag{20}$$

## D  BOP FOR EXPLAINABILITY - INCOMPREHENSIVENESS

**Classification**  Using the 0-1 loss function cost function defined for incomprehensiveness, the Minimal Group BoP is:

$$\begin{aligned} \gamma(h_0, h_p; \mathcal{D}) = \min_{\mathbf{s} \in \mathcal{S}} \big( &\Pr\left(h_p(\mathbf{X}, \mathbf{s}) \neq h_p(\mathbf{X}_{\setminus J}, \mathbf{s}_{\setminus J}) \mid \mathbf{S} = \mathbf{s}\right) \\ &- \Pr\left(h_0(\mathbf{X}) \neq h_0(\mathbf{X}_{\setminus J}) \mid \mathbf{S} = \mathbf{s}\right) \big), \quad \text{where} \quad \gamma \in [-1, 1]. \end{aligned}$$

**Regression** Using the square error loss function, the Minimal Group BoP for incomprehensiveness is:

$$\gamma(h_0, h_p; \mathcal{D}) = \min_{\mathbf{s} \in \mathcal{S}} \left( \mathbb{E} \left[ \|h_p(\mathbf{X}, \mathbf{s}) - h_p(\mathbf{X}_{\setminus J}, \mathbf{s}_{\setminus J})\|^2 \mid \mathbf{S} = \mathbf{s} \right] \right.$$
$$\left. - \mathbb{E} \left[ \|h_0(\mathbf{X}) - h_0(\mathbf{X}_{\setminus J})\|^2 \mid \mathbf{S} = \mathbf{s} \right] \right), \quad \text{where} \quad \gamma \in [-\infty, +\infty].$$

# E  PROOF OF THEOREMS ON LOWER BOUNDS FOR THE PROBABILITY OF ERROR

As in (Monteiro Paes et al., 2022), we will prove every theorem for the flipped hypothesis test defined as:

$$H_0: \quad \gamma(h_0, h_p; \mathcal{D}) \leq \epsilon \quad \Leftrightarrow \quad \text{Personalized } h_p \text{ performs worst: yields } \epsilon < 0 \text{ disadvantage}$$
$$H_1: \quad \gamma(h_0, h_p; \mathcal{D}) \geq 0 \quad \Leftrightarrow \quad \text{Personalized } h_p \text{ performs at least as good as generic } h_0.$$

where we emphasize that $\epsilon < 0$.

As shown in (Monteiro Paes et al., 2022), proving the bound for the original hypothesis test is equivalent to proving the bound for the flipped hypothesis test, since estimating $\gamma$ is as hard as estimating $-\gamma$. In every section that follows, $H_0, H_1$ refer to the flipped hypothesis test.

Here, we first prove a proposition that is valid for all of the cases that we consider in the next sections.

**Proposition 1.** *Consider $P_{\mathbf{X}, \mathbf{S}, Y}$ is a distribution of data, for which the generic model $h_0$ performs better, i.e., the true $\gamma$ is such that $\gamma(h_0, h_p, \mathcal{D}) < 0$, and $Q_{\mathbf{X}, \mathbf{S}, Y}$ is a distribution of data points for which the personalized model performs better, i.e., the true $\gamma$ is such that $\gamma(h_0, h_p, \mathcal{D}) > 0$. Consider a decision rule $\Psi$ that represents any hypothesis test. We have the following bound on the probability of error $P_e$:*

$$\min_{\Psi} \max_{\substack{P_0 \in H_0 \\ P_1 \in H_1}} P_e \geq 1 - TV(P \parallel Q),$$

*for any well-chosen $P \in H_0$ and any well-chosen $Q \in H_1$. Here TV refers to the total variation between probability distributions $P$ and $Q$.*

*Proof.* Consider $h_0$ and $h_p$ fixed. Take one decision rule $\Psi$ that represents any hypothesis test. Consider a dataset such that $H_0$ is true, i.e., $\mathcal{D} \sim P_0$ and a dataset such that $H_1$ is true, i.e., $\mathcal{D} \sim P_1$.

It might seem weird to use two datasets to compute the same quantity $P_e$, i.e., one dataset to compute the first term in $P_e$, and one dataset to compute the second term in $P_e$. However, this is just a reflection of the fact that the two terms in $P_e$ come from two different settings: $H_0$ true or $H_0$ false, which are disjoint events: in the same way that $H_0$ cannot be simultaneously true and false, yet each term in $P_e$ consider one or the other case; then we use one or the other dataset.

We have:

$$P_e = \Pr(\text{Rejecting } H_0 | H_0 \text{ true}) + \Pr(\text{Failing to reject } H_0 | H_1 \text{ true})$$
$$= \Pr(\Psi(h_0, h_p, \mathcal{D}, \epsilon) = 1 | \mathcal{D} \sim P_0) + \Pr(\Psi(h_0, h_p, \mathcal{D}, \epsilon) = 0 | \mathcal{D} \sim P_1)$$
$$= \Pr(\Psi(\mathcal{D}) = 1 | \mathcal{D} \sim P_0) + \Pr(\Psi(\mathcal{D}) = 0 | \mathcal{D} \sim P_1) \text{ simplifying notations}$$
$$= 1 - \Pr(\Psi(\mathcal{D}) = 0 | \mathcal{D} \sim P_0) + \Pr(\Psi(\mathcal{D}) = 0 | \mathcal{D} \sim P_1) \text{ complementary event}$$
$$= 1 - P_0(E_\Psi) + P_1(E_\Psi) \text{ writing } E_\Psi \text{ the event } \Psi(\mathcal{D}) = 0$$
$$= 1 - (P_0(E_\Psi) - P_1(E_\Psi))$$

Now, we will bound this quantity:

$$\min_{\Psi} \max_{\substack{P_0 \in H_0 \\ P_1 \in H_1}} P_e = \min_{\Psi} \max_{\substack{P_0 \in H_0 \\ P_1 \in H_1}} 1 - (P_0(E_\Psi) - P_1(E_\Psi))$$

$$\geq \max_{\substack{P_0 \in H_0 \\ P_1 \in H_1}} \min_{\Psi} \left[ 1 - (P_0(E_\Psi) - P_1(E_\Psi)) \right] \text{ using minmax inequality}$$

$$= \max_{\substack{P_0 \in H_0 \\ P_1 \in H_1}} \left[ 1 - \max_{\Psi}(P_0(E_\Psi) - P_1(E_\Psi)) \right] \text{ to get min over } \Psi, \text{ we want } (P_0(E_\Psi) - P_1(E_\Psi)) \text{ that is largest.}$$

$$\geq \max_{\substack{P_0 \in H_0 \\ P_1 \in H_1}} \left[ 1 - \max_{\text{events } A}(P_0(A) - P_1(A)) \right] \text{ because the max is now over all possible events } A$$

The maximization is broadened to consider all possible events $A$. This increases the set over which the maximum is taken. Because $\Psi$ is only a subset of all possible events, maximizing over all events $A$ (which includes $\Psi$) will result in a value that is at least as large as the maximum over $\Psi$. In other words, extending the set of possible events can only make the maximum greater or the same.

$$= \max_{\substack{P_0 \in H_0 \\ P_1 \in H_1}} \left[ 1 - TV(P_0 \parallel P_1) \right] \text{ by definition of the total variation (TV)}$$

$$= 1 - \min_{\substack{P_0 \in H_0 \\ P_1 \in H_1}} TV(P_0 \parallel P_1)$$

$$\geq 1 - TV(P \parallel Q) \text{ for any } P \in H_0 \text{ and } Q \in H_1.$$

This is true because the total variation distance $TV(P \parallel Q)$ for any particular pair $P$ and $Q$ cannot be smaller than the minimum total variation distance across all pairs. We recall that, by definition, the total variation of two probability distributions $P, Q$ is the largest possible difference between the probabilities that the two probability distributions can assign to the same event $A$. $\square$

Next, we prove a lemma that will be useful for the follow-up proofs.

**Lemma 3.** *Consider a random variable $a$ such that $\mathbb{E}[a] = 1$. Then:*

$$\mathbb{E}[(a-1)^2] = \mathbb{E}[a^2] - 1 \tag{21}$$

*Proof.* We have that:

$$\mathbb{E}[(a-1)^2] = \mathbb{E}[a^2 - 2a + 1]$$
$$= \mathbb{E}[a^2] - 2\mathbb{E}[a] + 1 \text{ (linearity of the expectation)}$$
$$= \mathbb{E}[a^2] - 2 + 1 (\mathbb{E}[a] = 1 \text{ by assumption})$$
$$= \mathbb{E}[a^2] - 1.$$

$\square$

### E.1 PROOF FOR CATEGORICAL BoP

Here, we redo the proof from Monteiro Paes (Monteiro Paes et al., 2022), to find a tighter bound.

**Theorem 4** (Lower bound for categorical individual BoP (Monteiro Paes et al., 2022)). *The lower bound writes:*

$$\min_{\Psi} \max_{\substack{P_{\mathbf{X}, \mathbf{S}, Y} \in H_0 \\ Q_{\mathbf{X}, \mathbf{S}, Y} \in H_1}} P_e \geq 1 - \frac{1}{2\sqrt{d}} \left(1 + 4\epsilon^2\right)^{m/2} \tag{22}$$

*where $P_{\mathbf{X}, \mathbf{S}, Y}$ is a distribution of data, for which the generic model $h_0$ performs better, i.e., the true $\gamma$ is such that $\gamma(h_0, h_p, \mathcal{D}) < 0$, and $Q_{\mathbf{X}, \mathbf{S}, Y}$ is a distribution of data points for which the personalized model performs better, i.e., the true $\gamma$ is such that $\gamma(h_0, h_p, \mathcal{D}) \geq \epsilon$. Dataset $\mathcal{D}$ is drawn from an unknown distribution and has $d$ groups where $d = 2^k$, with each group having $m = \lfloor N/d \rfloor$ samples.*

By recognizing that the variance of a Bernouilli distribution of parameter $p$ is $\sigma^2 = p(1-p) = \frac{1}{2}(1-\frac{1}{2}) = \frac{1}{4}$, we see that the lower bound can equivalently be written:

$$L = 1 - \frac{1}{2\sqrt{d}}\left(1 + 4\epsilon^2\right)^{m/2} = 1 - \frac{1}{2\sqrt{d}}\left(1 + \frac{\epsilon^2}{\sigma^2}\right)^{m/2} \tag{23}$$

This equivalent formulation is interesting to compare this bound to the bound obtained for the Gaussian case in the next section. In particular, we see that both bounds enjoy a very similar structure, where the key variable controlling the bound is $\frac{\epsilon}{\sigma}$ which is the minimum benefit of personalization $\epsilon$ at the scale of the variance of the benefits across groups.

*Proof.* By Proposition 1, we have:

$$\min_{\Psi} \max_{\substack{P_0 \in H_0 \\ P_1 \in H_1}} P_e \geq 1 - TV(P \parallel Q)$$

for any well-chosen $P \in H_0$ and any well-chosen $Q \in H_1$.

We will design two probability distributions $P, Q$ defined on the $N$ data points $(X_1, G_1, Y_1), ..., (X_N, G_N, Y_N)$ of the dataset $\mathcal{D}$ to compute an interesting right hand side term. An "interesting" right hand side term is a term that makes the lower bound as tight as possible, i.e., it relies on distributions $P, Q$ for which $TV(P \parallel Q)$ is small, i.e., probability distributions that are similar. To achieve this, we will first design the distribution $Q \in H_1$, and then propose $P$ as a very small modification of $Q$, just enough to allows it to verify $P \in H_0$.

Mathematically, $P, Q$ are distributions on the dataset $\mathcal{D}$, i.e., on $N$ i.i.d. realizations of the random variables $X, S, Y$ where $X$ is continuous, $S$ is categorical (binary), and $Y$ is binary (classification framework). Thus, we wish to design probability distributions on $(X_1, S_1, Y_1), ..., (X_N, S_N, Y_N)$.

However, we note that the dataset distribution is only meaningful in terms of how each triplet $(X_i, S_i, Y_i)$ impacts the value of the estimated BOP. Thus, we design probability distributions $P, Q$ on $N$ i.i.d. realizations of an auxiliary random variable $B$, with values in $\mathbb{R}$, defined as:

$$B = \ell(h_0(X), Y) - \ell(h_p(X, S), Y). \tag{24}$$

Intuitively, $B_i$ represents how much the triplet $(X_i, S_i, Y_i)$ contributes to the value of the BOP. $b_i > 0$ means that the personalized model provided a better prediction than the generic model on the triplet $(x_i, s_i, y_i)$ corresponding to the data point $i$.

In the case of classification, prediction or explainability approach, $\ell(h_0(X), Y)$ and $\ell(h_p(X, S), Y)$ are Bernouilli random variables, taking values in $\{0, 1\}$, while their difference $B$ is a categorical random variable taking values in $\{-1, 0, 1\}$.

Consider the event $b = (b_1, ..., b_N) \in \mathbb{R}^N$ of $N$ realizations of $B$. For simplicity in our computations, we divide this event into the $d$ groups, i.e., we write instead: $b_j = (b_j^{(1)}, ..., b_j^{(m)})$, since each group $j$ has $m$ samples. Thus, we have: $b = \{b_j^{(k)}\}_{j=1...d, k=1...m}$ indexed by $j, k$ where $j = 1...d$ is the group in which this element is, and $k = 1...m$ is the index of the element in that group.

In what follows, we denote $\text{Cat}(p_1, 1 - p_1 - p_2, p_2)$ the ternary categorical distribution, i.e., $\text{Cat}(p_1, 1 - p_1 - p_2, p_2) = -1$ with probability $p_1$, $\text{Cat}(p_1, 1 - p_1 - p_2, p_2) = 1$ with probability $p_2$, and $\text{Cat}(p_1, 1 - p_1 - p_2, p_2) = 0$ with probability $1 - p_1 - p_2$.

**Design $Q$.** Consider $p = \text{Cat}(\frac{1}{2}, 0, \frac{1}{2})$ a centered Categorical distribution, we propose the following distribution for $Q$:

$$Q_j(b_j) = \prod_{k=1}^{m} p(b_j^{(k)}), \text{ for every group } j = 1....d$$

$$Q(b) = \prod_{j=1}^{d} Q_j(b_j).$$

**Design $P$.**    Next, we design $P$ as a small modification of the distribution $Q$, that will just be enough to get $P \in H_0$. We recall that $P \in H_0$ means that $\gamma \leq \epsilon$ where $\epsilon < 0$ in the flipped hypothesis test. This means that, under $H_0$, there is one group that suffers a decrease of performance of $|\epsilon|$ because of the personalized model.

Given $p = \text{Cat}(\frac{1}{2}, 0, \frac{1}{2})$ a centered categorical distribution, and $p^\epsilon = \text{Cat}(\frac{1}{2}+\epsilon, 0, \frac{1}{2}-\epsilon)$ a categorical distribution with negative mean $\epsilon < 0$, we have:

$$P_j(b_j) = \prod_{k=1}^m p(b_j^{(k)}), \text{ for every group } j = 1....d,$$

$$P_j^\epsilon(b_j) = \prod_{k=1}^m p^\epsilon(b_j^{(k)}), \text{ for every group } j = 1....d,$$

$$P(b) = \frac{1}{d} \sum_{j=1}^d P_j^\epsilon(b_j) \prod_{j' \neq j} P_{j'}(b_{j'}).$$

**Compute total variation $TV(P \parallel Q)$.**    Given $P$ and $Q$, we can compute their total variation:

$$\text{TV}(P\|Q) = \frac{1}{2} \sum_{b_1,\ldots,b_d} |P(b_1,\ldots,ba_d) - Q(b_1,\ldots,b_d)| \text{ (TV for probability mass functions)}$$

$$= \frac{1}{2} \sum_{b_1,\ldots,b_d} \left| \frac{1}{d} \sum_{j=1}^d P_j^\epsilon(b_j) \prod_{j' \neq j} P_{j'}(b_{j'}) - \prod_{j=1}^d Q_j(b_j) \right| \text{ (definition of } P, Q)$$

$$= \frac{1}{2} \sum_{b_1,\ldots,b_d} \left| \frac{1}{d} \sum_{j=1}^d \frac{P_j^\epsilon(b_j)}{P_j(b_j)} \prod_{j'=1}^d P_{j'}(b_{j'}) - \prod_{j=1}^d Q_j(b_j) \right| \text{ (adding missing } j' = j)$$

$$= \frac{1}{2} \sum_{b_1,\ldots,b_d} \left| \frac{1}{d} \sum_{j=1}^d \frac{P_j^\epsilon(b_j)}{P_j(b_j)} \prod_{j'=1}^d Q_{j'}(b_{j'}) - \prod_{j=1}^d Q_j(b_j) \right| \text{ (} P_j = Q_j \text{ by construction)}$$

$$= \frac{1}{2}\mathbb{E}_Q \left[ \left| \frac{1}{d} \sum_{j=1}^d \frac{P_j^\epsilon(b_j)}{P_j(b_j)} - 1 \right| \right] \text{ (recognizing an expectation with respect to } Q)$$

$$= \frac{1}{2}\mathbb{E}_Q \left[ \left| \frac{1}{d} \sum_{j=1}^d \frac{\prod_{k=1}^m p^\epsilon(b_j^{(k)})}{\prod_{k=1}^m p(b_j^{(k)})} - 1 \right| \right] \text{ (definition of } P_j \text{ and } P_j^{(\epsilon)})$$

**Plug in the categorical assumption**    Under the assumption of a categorical distribution for the random variable $B$, we have:

$$\text{TV}(P\|Q) = \frac{1}{2}\mathbb{E}_Q \left[ \left| \frac{1}{d} \sum_{j=1}^d \prod_{k=1}^m (1+2\epsilon)^{\frac{1-b_j^k}{2}} (1-2\epsilon)^{\frac{1+b_j^k}{2}} - 1 \right| \right] \text{ (definition of } p \text{ and } p^{(\epsilon)})$$

$$= \frac{1}{2}\mathbb{E}_Q \left[ \left| \frac{1}{d} \sum_{j=1}^d \prod_{k=1}^m (1+2\epsilon)^{\hat{b}^k} (1-2\epsilon)^{1-\hat{b}^k} - 1 \right| \right] \text{ (Define } \hat{b}_j \triangleq (1 - b_j)/2, \text{ element wise)}$$

$$= \frac{1}{2}\mathbb{E}_Q \left[ \left| \frac{1}{d} \sum_{j=1}^d (1+2\epsilon)^{\sum_{k=1}^m \hat{b}^k} (1-2\epsilon)^{m-\sum_{k=1}^m \hat{b}^k} - 1 \right| \right] \text{ (property of power)}$$

Given that each $b_j$ is distributed as a Bernouilli distribution $Ber(1/2)$, we have that each entry of $\hat{b}_i$ is also distributed as a Bernoulli distribution with parameter $1/2$. Define $z_j \triangleq \sum_{k=1}^m \hat{b}_j^k$. By

definition, a binomial random variable $Bin(m,p)$ is a sum of $m$ Bernouilli random variables of probability $p$. Thus, as a sum of $m$ Bernouilli random variables $Ber(1/2)$, $z_j$ is distributed as a Binomial distribution $Bin(m, 1/2)$.

$$\text{TV}(P\|Q) = \frac{1}{2}\mathbb{E}\left[\left|\frac{1}{d}\sum_{j=1}^{d}(1+2\epsilon)^{z_j}(1-2\epsilon)^{m-z_j} - 1\right|\right]$$

$$\leq \frac{1}{2}\mathbb{E}\left[\left(\frac{1}{d}\sum_{j=1}^{d}(1+2\epsilon)^{z_j}(1-2\epsilon)^{m-z_j} - 1\right)^2\right]^{1/2} \quad \text{(by Cauchy-Schwarz: } \mathbb{E}[|X|] \leq \sqrt{\mathbb{E}[X^2]} \text{ )}$$

This last inequality is where our proof differs from (Monteiro Paes et al., 2022). Indeed, while the authors drop the factor $\frac{1}{2}$, by contrast, here, we choose to keep it.

**Auxiliary computation to apply Lemma 3** Next, we will apply Lemma 3. For this, we need to prove that the expectation of the first term is 1. We perform this auxiliary computation here. We recall that the moment generating function (MGF) of a binomial $Bin(m,p)$ is, by definition, $M(t) = (q + pe^t)^m$ where $q = 1 - p$. We have that:

$$\mathbb{E}\left[\frac{1}{d}\sum_{i=1}^{d}(1+2\epsilon)^{z_i}(1-2\epsilon)^{m-z_i}\right] = \frac{1}{d}(1-2\epsilon)^m\sum_{i=1}^{d}\mathbb{E}\left[\left(\frac{1+2\epsilon}{1-2\epsilon}\right)^{z_i}\right] \quad \text{(extracting } (1-2\epsilon)^m \text{ out of the sum)}$$

$$= \frac{1}{d}(1-2\epsilon)^m\sum_{i=1}^{d}\mathbb{E}\left[e^{z_i\ln\left(\frac{1+2\epsilon}{1-2\epsilon}\right)}\right] \quad \text{(definition of power, we recognize a MGF } \mathbb{E}[e^{zt}])$$

$$= \frac{1}{d}(1-2\epsilon)^m\sum_{i=1}^{d}\left(\frac{1}{2} + \frac{1}{2}\cdot\frac{1+2\epsilon}{1-2\epsilon}\right)^m \quad \text{(MGF of } Bin(m,\frac{1}{2}) \text{ for } t = \ln\left(\frac{1+2\epsilon}{1-2\epsilon}\right))$$

$$= \frac{1}{d}(1-2\epsilon)^m\sum_{i=1}^{d}\frac{1}{2^m}\left(1 + \frac{1+2\epsilon}{1-2\epsilon}\right)^m \quad \text{(extracting } \frac{1}{2} \text{ out of the power)}$$

$$= \frac{1}{d}(1-2\epsilon)^m\sum_{i=1}^{d}\frac{1}{2^m}\left(\frac{1-2\epsilon}{1-2\epsilon} + \frac{1+2\epsilon}{1-2\epsilon}\right)^m$$

$$= \frac{1}{d}(1-2\epsilon)^m\sum_{i=1}^{d}\frac{1}{2^m}\left(\frac{2}{1-2\epsilon}\right)^m$$

$$= \frac{1}{d}(1-2\epsilon)^m\sum_{i=1}^{d}\left(\frac{1}{1-2\epsilon}\right)^m \quad \text{(simplifying the terms with } 2^m)$$

$$= \frac{1}{d}(1-2\epsilon)^m d\left(\frac{1}{1-2\epsilon}\right)^m \quad \text{(term in the sum does not depend on } j)$$

$$= 1.$$

**Continue by applying Lemma 3.** This auxiliary computation shows that we meet the assumption of Lemma 3. Thus, we continue the computation of the lower bound of the TV by applying Lemma 3.

$\mathrm{TV}(P\|Q)$

$$\leq \frac{1}{2}\mathbb{E}\left[\left(\frac{1}{d}\sum_{j=1}^{d}(1+2\epsilon)^{z_j}(1-2\epsilon)^{m-z_j}-1\right)^2\right]^{1/2}$$

$$= \frac{1}{2}\mathbb{E}\left[\left(\frac{1}{d}\sum_{i=1}^{d}(1+2\epsilon)^{z_i}(1-2\epsilon)^{m-z_i}\right)^2-1\right]^{1/2} \quad \text{(applying Lemma 3)}$$

$$= \frac{1}{2}\mathbb{E}\left[\left(\frac{1}{d^2}\sum_{i,j=1}^{d}(1+2\epsilon)^{z_i}(1-2\epsilon)^{m-z_i}(1+2\epsilon)^{z_j}(1-2\epsilon)^{m-z_j}\right)-1\right]^{1/2} \quad \text{(expanding square of the sum)}$$

**Expand double sum.** To continue, we will expand the double sum on indices $i, j$ into two parts: a part where $i = j$ and a part where $i \neq j$. In the latter, the random variables $z_i$ and $z_j$, for $i \neq j$, are independent. We recall here that they are Binomial random variables $Bin(m, \frac{1}{2})$. As the sum of independent Binomial random variables, of same probability of success (her: $p = \frac{1}{2}$), is also a Binomial random variable. Here, we will have: $z_i + z_j \sim Bin(2m, \frac{1}{2})$. We continue the computations:

$\mathrm{TV}(P\|Q)$

$$\leq \frac{1}{2}\mathbb{E}\left[\frac{1}{d^2}\sum_{i=1}^{d}\left((1+2\epsilon)^{z_i}(1-2\epsilon)^{m-z_i}\right)^2+\left(\frac{1}{d^2}\sum_{\substack{i,j=1\\i\neq j}}^{d}(1+2\epsilon)^{z_i}(1-2\epsilon)^{m-z_i}(1+2\epsilon)^{z_j}(1-2\epsilon)^{m-z_j}\right)-1\right]^{1/2}$$

$$= \frac{1}{2}\mathbb{E}\left[\frac{1}{d^2}\sum_{i=1}^{d}\left((1+2\epsilon)^{z_i}(1-2\epsilon)^{m-z_i}\right)^2+\frac{1}{d^2}\sum_{\substack{i,j=1\\i\neq j}}^{d}(1+2\epsilon)^{z_i+z_j}(1-2\epsilon)^{2m-z_i-z_j}-1\right]^{1/2} \quad \text{(property of power)}$$

$$= \frac{1}{2}\left(\frac{1}{d^2}\sum_{i=1}^{d}\mathbb{E}\left[\left((1+2\epsilon)^{z_i}(1-2\epsilon)^{m-z_i}\right)^2\right]+\frac{1}{d^2}\sum_{\substack{i,j=1\\i\neq j}}^{d}\mathbb{E}\left[(1+2\epsilon)^{z_i+z_j}(1-2\epsilon)^{2m-z_i-z_j}\right]-1\right)^{1/2}$$

(linearity of the expectation)

$$= \frac{1}{2}\left(\frac{1}{d^2}\sum_{i=1}^{d}\mathbb{E}\left[\left((1+2\epsilon)^{z_i}(1-2\epsilon)^{m-z_i}\right)^2\right]+\frac{1}{d^2}\sum_{\substack{i,j=1\\i\neq j}}^{d}\mathbb{E}\left[(1+2\epsilon)^{\tilde{z}}(1-2\epsilon)^{2m-\tilde{z}}\right]-1\right)^{1/2}$$

where we define the sum $z_i + z_j$ of two independent $Bin(m, \frac{1}{2})$ as the new Binomial variable: $\tilde{z} \sim Bin(2m, \frac{1}{2})$). Here, we can apply the result of the auxiliary computation above, to see that the expectation on $\tilde{z}$ is equal to 1.

Thus, we get:

$\text{TV}(P\|Q)$

$$\leq \frac{1}{2}\left(\frac{1}{d^2}\sum_{i=1}^{d}\mathbb{E}\left[\left((1+2\epsilon)^{z_i}(1-2\epsilon)^{m-z_i}\right)^2\right] + \frac{1}{d^2}\sum_{\substack{i,j=1\\i\neq j}}^{d}1 - 1\right)^{1/2}$$

$$= \frac{1}{2}\left(\mathbb{E}\left[\frac{1}{d^2}\sum_{i=1}^{d}\left((1+2\epsilon)^{z_i}(1-2\epsilon)^{m-z_i}\right)^2\right] + \frac{d(d-1)}{d^2} - 1\right)^{1/2} \quad \text{(Counting the 1s in the sum)}$$

$$= \frac{1}{2}\left(\mathbb{E}\left[\frac{1}{d^2}\sum_{i=1}^{d}\left((1+2\epsilon)^{z_i}(1-2\epsilon)^{m-z_i}\right)^2\right] + 1 - \frac{1}{d} - 1\right)^{1/2} \quad \text{(simplifying)}$$

$$= \frac{1}{2}\left(\mathbb{E}\left[\frac{1}{d^2}\sum_{i=1}^{d}\left((1+2\epsilon)^{z_i}(1-2\epsilon)^{m-z_i}\right)^2\right] - \frac{1}{d}\right)^{1/2} \quad \text{(simplifying)}$$

$$= \frac{1}{2\sqrt{d}}\left(\mathbb{E}\left[\frac{1}{d}\sum_{i=1}^{d}\left((1+2\epsilon)^{z_i}(1-2\epsilon)^{m-z_i}\right)^2\right] - 1\right)^{1/2} \quad \text{(extracting } \frac{1}{d}\text{)}$$

$$\leq \frac{1}{2\sqrt{d}}\left(\mathbb{E}\left[\frac{1}{d}\sum_{i=1}^{d}\left((1+2\epsilon)^{z_i}(1-2\epsilon)^{m-z_i}\right)^2\right]\right)^{1/2} \quad \text{(because } \sqrt{a-1}\leq\sqrt{a} \text{ as } \sqrt{()} \text{ is monotonic increasing)}$$

$$= \frac{1}{2\sqrt{d}}\left(\frac{1}{d}\sum_{i=1}^{d}\mathbb{E}\left[\left((1+2\epsilon)^{z_i}(1-2\epsilon)^{m-z_i}\right)^2\right]\right)^{1/2} \quad \text{(linearity of expectation)}$$

$$= \frac{1}{2\sqrt{d}}\left(\frac{1}{d}d.\mathbb{E}\left[\left((1+2\epsilon)^{z_1}(1-2\epsilon)^{m-z_1}\right)^2\right]\right)^{1/2} \quad \text{(the } z_i\text{'s are identically distributed)}$$

$$= \frac{1}{2\sqrt{d}}\left(\mathbb{E}\left[\left((1+2\epsilon)^{z_1}(1-2\epsilon)^{m-z_1}\right)^2\right]\right)^{1/2} \quad \text{(simplifying)}$$

$$= \frac{1}{2}\frac{1}{\sqrt{d}}(1-2\epsilon)^m\mathbb{E}\left[\left|(1+2\epsilon)^{z_1}(1-2\epsilon)^{-z_1}\right|^2\right]^{1/2} \quad \text{(extracting } (1-2\epsilon)^m\text{)}$$

$$= \frac{1}{2}\frac{1}{\sqrt{d}}(1-2\epsilon)^m\mathbb{E}\left[\left(\frac{1+2\epsilon}{1-2\epsilon}\right)^{2z_1}\right]^{1/2} \quad \text{(property of power)}$$

$$= \frac{1}{2}\frac{1}{\sqrt{d}}(1-2\epsilon)^m\mathbb{E}\left[\exp\left(2z_1\ln\left(\frac{1+2\epsilon}{1-2\epsilon}\right)\right)\right]^{1/2} \quad \text{(property of power)}$$

$$= \frac{1}{2}\frac{1}{\sqrt{d}}(1-2\epsilon)^m\left(M_{\text{Bin}(m,1/2)}\left(2\ln\left(\frac{1+2\epsilon}{1-2\epsilon}\right)\right)\right)^{1/2} \quad \text{(definition of MGF as } \mathbb{E}[\exp(zt)] \text{ for } t=2\ln\left(\frac{1+2\epsilon}{1-2\epsilon}\right)\text{)}$$

$$= \frac{1}{2}\frac{1}{\sqrt{d}}\left(1+4\epsilon^2\right)^{m/2} \quad \text{(MGF of a Binomial random variable)}$$

Consequently, we obtain:

$$\min_{\Psi}\max_{\substack{P_0\in H_0\\P_1\in H_1}}P_e \geq 1 - \text{TV}(P\|Q)$$

$$\Rightarrow \min_{\Psi}\max_{\substack{P_{P'}\in H_0\\P_1\in H_1}}P_e \geq 1 - \frac{1}{2\sqrt{d}}\left(1+4\epsilon^2\right)^{m/2}$$

which is a slightly different bound than (Monteiro Paes et al., 2022) due to the $\frac{1}{2}$ that we kept. $\qquad\square$

## E.2 PROOF FOR GAUSSIAN BOP

Here, we do the proof for a real-valued cost function, assuming that the BoP is a normal variable with a second moment bounded by $\sigma^2$.

**Theorem 5** (Lower bound for real-valued cost function). *The lower bound writes:*

$$\min_{\Psi} \max_{\substack{P_{\mathbf{X},\mathbf{s},Y} \in H_0 \\ Q_{\mathbf{X},\mathbf{s},Y} \in H_1}} P_e \geq 1 - \frac{1}{2\sqrt{d}} \exp\left(\frac{\epsilon^2}{\sigma^2}\right)^{m/2}$$

*where $P_{\mathbf{X},\mathbf{s},Y}$ is a distribution of data, for which the generic model $h_0$ performs better, i.e., the true $\gamma$ is such that $\gamma(h_0, h_p, \mathcal{D}) < 0$, and $Q_{\mathbf{X},\mathbf{s},Y}$ is a distribution of data points for which the personalized model performs better, i.e., the true $\gamma$ is such that $\gamma(h_0, h_p, \mathcal{D}) > 0$.*

For a centered Gaussian random variable $X$ of variance $s^2$, the MGF takes the form $M_X(t) = \exp(\frac{1}{2}s^2 t^2)$. Thus, the lower bound writes:

$$L = 1 - \frac{1}{2\sqrt{d}} \exp\left(\frac{\epsilon^2}{\sigma^2}\right)^{m/2} = 1 - \frac{1}{2\sqrt{d}} M_X\left(\frac{\epsilon\sqrt{2}}{\sigma^2}\right)^{m/2}. \tag{25}$$

*Proof.* By Proposition 1, we have that:

$$\min_{\Psi} \max_{\substack{P_0 \in H_0 \\ P_1 \in H_1}} P_e \geq 1 - TV(P \parallel Q)$$

for any well-chosen $P \in H_0$ and any well-chosen $Q \in H_1$. We will design two probability distributions $P, Q$ defined on the $N$ data points $(X_1, G_1, Y_1), ..., (X_N, G_N, Y_N)$ of the dataset $\mathcal{D}$ to compute an interesting right hand side term. An "interesting" right hand side term is a term that makes the lower bound as tight as possible, i.e., it relies on distributions $P, Q$ for which $TV(P \parallel Q)$ is small, i.e., probability distributions that are similar. To achieve this, we will first design the distribution $Q \in H_1$, and then propose $P$ as a very small modification of $Q$, just enough to allows it to verify $P \in H_0$.

Mathematically, $P, Q$ are distributions on the dataset $\mathcal{D}$, i.e., on $N$ i.i.d. realizations of the random variables $X, S, Y$ where $X$ is continuous, $S$ is categorical, and $Y$ is continuous (regression framework). Thus, we wish to design probability distributions on $(X_1, S_1, Y_1), ..., (X_N, S_N, Y_N)$.

However, we note that the dataset distribution is only meaningful in terms of how each triplet $(X_i, S_i, Y_i)$ impacts the value of the estimated BOP. Thus, we design probability distributions $P, Q$ on $n$ i.i.d. realizations of an auxiliary random variable $B$, with values in $\mathbb{R}$, defined as:

$$B = \ell(h_0(X), Y) - \ell(h_p(X, S), Y). \tag{26}$$

Intuitively, $B_i$ represents how much the triplet $(X_i, S_i, Y_i)$ contributes to the value of the BOP. $b_i > 0$ means that the personalized model provided a better prediction than the generic model on the triplet $(x_i, s_i, y_i)$ corresponding to the data point $i$.

Consider the event $b = (b_1, ..., b_N) \in \mathbb{R}^N$ of $N$ realizations of $B$. For simplicity in our computations, we divide this event into the $d$ groups, i.e., we write instead: $b_j = (b_j^{(1)}, ..., b_j^{(m)})$, since each group $j$ has $m$ samples. Thus, we have: $b = \{b_j^{(k)}\}_{j=1...d, k=1...m}$ indexed by $j, k$ where $j = 1...d$ is the group in which this element is, and $k = 1...m$ is the index of the element in that group.

**Design $Q$.** Next, we design a distribution $Q$ on this set of events that will (barely) verify $H_1$, i.e., such that the expectation of $B$ according to $Q$ will give $\gamma = 0$. We recall that $\gamma = 0$ means that the minimum benefit across groups is 0, implying that there might be some groups that have a $> 0$ benefit.

Given $p = \mathcal{N}(0, \sigma^2)$ a centered Gaussian distribution, we propose the following distribution for $Q$

$$Q_j(b_j) = \prod_{k=1}^{m} p(b_j^{(k)}), \text{ for every group } j = 1....d$$

$$Q(b) = \prod_{j=1}^{d} Q_j(b_j).$$

We verify that we have designed $Q$ correctly, i.e., we verify that $Q \in H_1$. When the dataset is distributed according to $Q$, we have:

$$
\begin{aligned}
\gamma &= \min_{s \in S} C_s(h_0, s) - C_s(h_p, s) \\
&= \min_{s \in S} \mathbb{E}_Q[\ell(h_0(\mathbf{X}), Y) \mid \mathbf{S} = \mathbf{s}] - \mathbb{E}_Q[\ell(h_p(\mathbf{X}), Y) \mid \mathbf{S} = \mathbf{s}] \text{ (by definition of group cost)} \\
&= \min_{s \in S} \mathbb{E}_Q[\ell(h_0(\mathbf{X}), Y) - \ell(h_p(\mathbf{X}), Y) \mid \mathbf{S} = \mathbf{s}] \text{ (by linearity of expectation)} \\
&= \min_{s \in S} \mathbb{E}_Q[B \mid \mathbf{S} = \mathbf{s}] \text{ (by definition of random variable } B) \\
&= \min_{s \in S} 0 \text{ (by definition of the probability distribution on } B) \\
&= 0.
\end{aligned}
$$

Thus, we find that $\gamma = 0$ which means that $\gamma \geq 0$, i.e., $Q \in H_1$.

**Design $P$.** Next, we design $P$ as a small modification of the distribution $Q$, that will just be enough to get $P \in H_0$. We recall that $P \in H_0$ means that $\gamma \leq \epsilon$ where $\epsilon < 0$ in the flipped hypothesis test. This means that, under $H_0$, there is one group that suffers a decrease of performance of $|\epsilon|$ because of the personalized model.

Given $p = \mathcal{N}(0, \sigma^2)$ a centered Gaussian distribution, and $p^\epsilon = \mathcal{N}(\epsilon, \sigma^2)$ a Gaussian distribution of same variance but negative mean $\epsilon < 0$, we have:

$$
P_j(b_j) = \prod_{k=1}^m p(b_j^{(k)}), \text{ for every group } j = 1....d,
$$

$$
P_j^\epsilon(b_j) = \prod_{k=1}^m p^\epsilon(b_j^{(k)}), \text{ for every group } j = 1....d,
$$

$$
P(b) = \frac{1}{d} \sum_{j=1}^d P_j^\epsilon(b_j) \prod_{j' \neq j} P_{j'}(b_{j'}).
$$

Intuitively, this distribution represents the fact that there is one group for which the personalized model worsen performances by $|\epsilon|$. We assume that this group can be either group 1, or group 2, etc, or group $d$, and consider these to be disjoint events: i.e., exactly only one group suffers the $|\epsilon|$ performance decrease. We take the union of these disjoint events and sum of probabilities using the Partition Theorem (Law of Total Probability) in the definition of $P$ above.

We verify that we have designed $P$ correctly, i.e., we verify that $P \in H_0$. When the dataset is distributed according to $P$, we have:

$$
\begin{aligned}
\gamma &= \min_{s \in S} C_s(h_0, s) - C_s(h_p, s) \\
&= \min_{s \in S} \mathbb{E}_P[B \mid \mathbf{S} = \mathbf{s}] \text{ (same computations as for } Q \in H_1) \\
&= \min(\epsilon, 0, ..., 0) \text{ (since exactly one group has mean } \epsilon) \\
&= \epsilon \text{ (since } \epsilon < 0).
\end{aligned}
$$

Thus, we find that $\gamma = \epsilon$ which means that $\gamma \leq 0$, i.e., $P \in H_0$.

**Compute total variation $TV(P \parallel Q)$.** We have verified that $Q \in H_1$ and that $P \in H_0$. We use these probability distributions to compute the lower bound to $P_e$. First, we compute their total variation:

$$TV(P \parallel Q) = \frac{1}{2} \int_{b_1,...,b_j} |P(b_1,...,b_j) - Q(b_1,...,b_j)| \, db_1...db_j \text{ (TV for probability density functions)}$$

$$= \frac{1}{2} \int_{b_1,...,b_j} \left| \frac{1}{d} \sum_{j=1}^{d} P_j^{\epsilon}(b_j) \prod_{j' \neq j} P_{j'}(b_{j'}) - \prod_{j=1}^{d} Q_j(b_j) \right| db_1...db_j \text{ (definition of } P,Q)$$

$$= \frac{1}{2} \int_{b_1,...,b_j} \left| \frac{1}{d} \sum_{j=1}^{d} \frac{P_j^{\epsilon}(b_j)}{P_j(b_j)} \prod_{j'=1}^{d} P_{j'}(b_{j'}) - \prod_{j=1}^{d} Q_j(b_j) \right| db_1...db_j \text{ (adding missing } j'=j)$$

$$= \frac{1}{2} \int_{b_1,...,b_j} \left| \frac{1}{d} \sum_{j=1}^{d} \frac{P_j^{\epsilon}(b_j)}{P_j(b_j)} \prod_{j'=1}^{d} Q_{j'}(b_{j'}) - \prod_{j=1}^{d} Q_j(b_j) \right| db_1...db_j \ (P_j = Q_j \text{ by construction})$$

$$= \frac{1}{2} \int_{b_1,...,b_j} \prod_{j=1}^{d} Q_j(b_j) \left| \frac{1}{d} \sum_{j=1}^{d} \frac{P_j^{\epsilon}(b_j)}{P_j(b_j)} - 1 \right| db_1...db_j \text{ (extracting the product)}$$

$$= \frac{1}{2} \mathbb{E}_Q \left[ \left| \frac{1}{d} \sum_{j=1}^{d} \frac{P_j^{\epsilon}(b_j)}{P_j(b_j)} - 1 \right| \right] \text{ (recognizing an expectation with respect to } Q)$$

$$= \frac{1}{2} \mathbb{E}_Q \left[ \left| \frac{1}{d} \sum_{j=1}^{d} \frac{\prod_{k=1}^{m} p^{\epsilon}(b_j^{(k)})}{\prod_{k=1}^{m} p(b_j^{(k)})} - 1 \right| \right] \text{ (definition of } P_j \text{ and } P_j^{(\epsilon)})$$

**Plug in the Gaussian assumption.** Under the assumption of Gaussianity of the random variable $B$, we continue the computations as:

$$TV(P \parallel Q) = \frac{1}{2}\mathbb{E}_Q\left[\left|\frac{1}{d}\sum_{j=1}^{d}\frac{\prod_{k=1}^{m}\exp(-\frac{\|b_j^{(k)}-\epsilon\|^2}{2\sigma^2})}{\prod_{k=1}^{m}\exp(-\frac{\|b_j^{(k)}\|^2}{2\sigma^2})}-1\right|\right] \quad \text{(definition of } p \text{ and } p^{(\epsilon)})$$

$$= \frac{1}{2}\mathbb{E}_Q\left[\left|\frac{1}{d}\sum_{j=1}^{d}\exp\left(-\frac{\sum_{k=1}^{m}\left(\|b_j^{(k)}-\epsilon\|^2-\|b_j^{(k)}\|^2\right)}{2\sigma^2}\right)-1\right|\right] \quad \text{(property of exp)}$$

$$= \frac{1}{2}\mathbb{E}_Q\left[\left|\frac{1}{d}\sum_{j=1}^{d}\exp\left(-\frac{\|b_j-\bar\epsilon\|_2^2-\|b_j\|_2^2}{2\sigma^2}\right)-1\right|\right] \quad \text{(with } \bar\epsilon = (\epsilon,..,\epsilon) \in \mathbb{R}^m)$$

$$= \frac{1}{2}\mathbb{E}_Q\left[\left|\frac{1}{d}\sum_{j=1}^{d}\exp\left(-\frac{\|b_j\|_2^2-2<b_j,\bar\epsilon>+\|\bar\epsilon\|_2^2-\|b_j\|_2^2}{2\sigma^2}\right)-1\right|\right] \quad \text{(expansion of } \|\|_2^2)$$

$$= \frac{1}{2}\mathbb{E}_Q\left[\left|\frac{1}{d}\sum_{j=1}^{d}\exp\left(-\frac{-2<b_j,\bar\epsilon>+\|\bar\epsilon\|_2^2}{2\sigma^2}\right)-1\right|\right] \quad \text{(simplifying)}$$

$$= \frac{1}{2}\mathbb{E}_Q\left[\left|\frac{1}{d}\exp\left(-\frac{\|\bar\epsilon\|_2^2}{2\sigma^2}\right)\sum_{j=1}^{d}\exp\left(\frac{-2<b_j,\bar\epsilon>}{2\sigma^2}\right)-1\right|\right] \quad \text{(since } \bar\epsilon \text{ does not depend on } j)$$

$$= \frac{1}{2}\mathbb{E}_Q\left[\left|\frac{1}{d}\exp\left(-\frac{m\epsilon^2}{2\sigma^2}\right)\sum_{j=1}^{d}\exp\left(\frac{2<b_j,\bar\epsilon>}{2\sigma^2}\right)-1\right|\right] \quad \text{(definition of } \bar\epsilon = \epsilon.1_m)$$

$$= \frac{1}{2}\mathbb{E}_Q\left[\left|\frac{1}{d}\exp\left(-\frac{m\epsilon^2}{2\sigma^2}\right)\sum_{j=1}^{d}\exp\left(\frac{<b_j,\bar\epsilon>}{\sigma^2}\right)-1\right|\right]$$

$$= \frac{1}{2}\mathbb{E}_Q\left[\left|\frac{1}{d}\exp\left(-\frac{m\epsilon^2}{2\sigma^2}\right)\sum_{j=1}^{d}\exp\left(\frac{\epsilon\sum_{k=1}^{m}b_j^{(k)}}{\sigma^2}\right)-1\right|\right] \quad \text{(because } \bar\epsilon = (\epsilon,..,\epsilon)).$$

Next, we define the auxiliary random variables $z_j = \frac{\epsilon}{\sigma^2}\sum_{k=1}^{m}b_j^{(k)}$. The sum of independent, identically distributed $N(0,\sigma^2)$ Gaussian random variables $\sum_{k=1}^{m}b_j^{(k)}$ is itself a Gaussian random variable distributed as $N(0,m\sigma^2)$. Scaling this random variable by $\frac{\epsilon}{\sigma^2}$ gives a random variable $z_j$ distributed as $\mathcal{N}(0,\frac{\epsilon^2}{\sigma^4}m\sigma^2) = \mathcal{N}(0,\frac{m\epsilon^2}{\sigma^2})$. Thus, we get:

$$TV(P \parallel Q) = \frac{1}{2}\mathbb{E}_Q\left[\left|\frac{1}{d}\exp\left(-\frac{m\epsilon^2}{2\sigma^2}\right)\sum_{j=1}^{d}\exp\left(\frac{\epsilon\sum_{k=1}^{m}b_j^{(k)}}{\sigma^2}\right)-1\right|\right]$$

$$= \frac{1}{2}\mathbb{E}\left[\left|\frac{1}{d}\exp\left(-\frac{m\epsilon^2}{2\sigma^2}\right)\sum_{j=1}^{d}\exp z_j-1\right|\right] \quad \text{(where } z_j \sim N(0,\frac{m\epsilon^2}{\sigma^2}))$$

$$\leq \frac{1}{2}\mathbb{E}\left[\left|\frac{1}{d}\exp\left(-\frac{m\epsilon^2}{2\sigma^2}\right)\sum_{j=1}^{d}\exp z_j-1\right|^2\right]^{\frac{1}{2}} \quad \text{(by Cauchy-Schwartz: } \mathbb{E}[|X|] \leq \sqrt{\mathbb{E}[X^2]})$$

**Auxiliary computation to apply Lemma 3** Next, we will apply Lemma 3. For this, we need to prove that the expectation of the first term is 1. We perform this auxiliary computation here. We

recall that the moment generating function (MGF) of a centered Gaussian random variable $X$ of variance $s^2$ is $M_X(t) = \exp(\frac{1}{2}s^2 t^2)$. We have that:

$$\mathbb{E}\left[\frac{1}{d}\exp\left(-\frac{m\epsilon^2}{2\sigma^2}\right)\sum_{j=1}^{d}\exp z_j\right] = \frac{1}{d}\exp\left(-\frac{m\epsilon^2}{2\sigma^2}\right)\sum_{j=1}^{d}\mathbb{E}\left[\exp z_j\right] \text{ (linearity of expectation)}$$

$$= \frac{1}{d}\exp\left(-\frac{m\epsilon^2}{2\sigma^2}\right)\sum_{j=1}^{d}\exp(\frac{1}{2}\frac{m\epsilon^2}{\sigma^2}) \text{ (MGF of centered Gaussian)}$$

$$= \frac{1}{d}\exp\left(-\frac{m\epsilon^2}{2\sigma^2}\right) d.\exp(\frac{1}{2}\frac{m\epsilon^2}{\sigma^2}) \text{ (term in the sum is independent of } i)$$

$$= 1.$$

**Continue by applying Lemma 3.** This auxiliary computation shows that we meet the assumption of Lemma 3. Thus, we continue the computation of the lower bound of the TV by applying Lemma 3.

$$TV(P \parallel Q)$$

$$\leq \frac{1}{2}\mathbb{E}\left[\left(\frac{1}{d}\exp\left(-\frac{m\epsilon^2}{2\sigma^2}\right)\sum_{j=1}^{d}\exp z_j\right)^2 - 1\right]^{\frac{1}{2}} \text{ (applying Lemma 3)}$$

$$= \frac{1}{2}\mathbb{E}\left[\left(\frac{1}{d^2}\exp\left(-\frac{m\epsilon^2}{\sigma^2}\right)\sum_{j,j'=1}^{d}\exp z_j \exp z_{j'}\right) - 1\right]^{\frac{1}{2}} \text{ (expanding the square of the sum)}$$

$$= \frac{1}{2}\mathbb{E}\left[\left(\frac{1}{d^2}\exp\left(-\frac{m\epsilon^2}{\sigma^2}\right)\sum_{j,j'=1}^{d}\exp\left(z_j + z_{j'}\right)\right) - 1\right]^{\frac{1}{2}} \text{ (property of exp)}$$

$$= \frac{1}{2}\mathbb{E}\left[\left(\frac{1}{d^2}\exp\left(-\frac{m\epsilon^2}{\sigma^2}\right)\sum_{j=1}^{d}\exp\left(2z_j\right)\right) + \left(\frac{1}{d^2}\exp\left(-\frac{m\epsilon^2}{\sigma^2}\right)\sum_{j,j'=1,j'\neq j}^{d}\exp\left(z_j + z_{j'}\right)\right) - 1\right]^{\frac{1}{2}}$$

where we split the double sum to get independent variables in the second term.

We get by linearity of the expectation, $\mathbb{E}[aX + bY] = a\mathbb{E}[X] + b\mathbb{E}[Y]$:

$$TV(P \parallel Q)$$

$$\leq \frac{1}{2}\mathbb{E}\left[\left(\frac{1}{d^2}\exp\left(-\frac{m\epsilon^2}{\sigma^2}\right)\sum_{j=1}^{d}\exp\left(2z_j\right)\right) + \left(\frac{1}{d^2}\exp\left(-\frac{m\epsilon^2}{\sigma^2}\right)\sum_{j,j'=1,j'\neq j}^{d}\exp\left(z_j + z_{j'}\right)\right) - 1\right]^{\frac{1}{2}}$$

$$= \frac{1}{2}\left[\left(\frac{1}{d^2}\exp\left(-\frac{m\epsilon^2}{\sigma^2}\right)\sum_{j=1}^{d}\mathbb{E}[\exp\left(2z_j\right)]\right) + \left(\frac{1}{d^2}\exp\left(-\frac{m\epsilon^2}{\sigma^2}\right)\sum_{j,j'=1,j'\neq j}^{d}\mathbb{E}[\exp\left(z_j + z_{j'}\right)]\right) - 1\right]^{\frac{1}{2}}$$

Here, $2z_j \sim \mathcal{N}(0, 4\frac{m\epsilon^2}{\sigma^2})$ and independent sum is $z_j + z_{j'} \sim \mathcal{N}(0, 2\frac{m\epsilon^2}{\sigma^2})$. In both cases, we recognize the moment generating function (MGF) of a random variable, defined as $M_X(t) = \mathbb{E}[\exp(tX)]$ evaluated at $t = 1$. For a centered Gaussian random variable $X$ of variance $s^2$, the MGF takes the

form $M_X(t) = \exp(\frac{1}{2}s^2 t^2)$. Applying this to our two random variables $2z_j$ and $z_j + z_{j'}$, we get:

$$TV(P \parallel Q)$$

$$\leq \frac{1}{2}\left[\left(\frac{1}{d^2}\exp\left(-\frac{m\epsilon^2}{\sigma^2}\right)\sum_{j=1}^{d}\mathbb{E}[\exp\left(2z_j\right)]\right) + \left(\frac{1}{d^2}\exp\left(-\frac{m\epsilon^2}{\sigma^2}\right)\sum_{j,j'=1,j'\neq j}^{d}\mathbb{E}[\exp\left(z_j + z_{j'}\right)]\right) - 1\right]^{\frac{1}{2}}$$

$$= \frac{1}{2}\left[\left(\frac{1}{d^2}\exp\left(-\frac{m\epsilon^2}{\sigma^2}\right)\sum_{j=1}^{d}\exp(\frac{1}{2}4\frac{m\epsilon^2}{\sigma^2})\right) + \left(\frac{1}{d^2}\exp\left(-\frac{m\epsilon^2}{\sigma^2}\right)\sum_{j,j'=1,j'\neq j}^{d}\exp(\frac{1}{2}2\frac{m\epsilon^2}{\sigma^2})\right) - 1\right]^{\frac{1}{2}}$$

$$= \frac{1}{2}\left[\left(\frac{1}{d^2}\exp\left(-\frac{m\epsilon^2}{\sigma^2}\right)\sum_{j=1}^{d}\exp(2\frac{m\epsilon^2}{\sigma^2})\right) + \left(\frac{1}{d^2}\exp\left(-\frac{m\epsilon^2}{\sigma^2}\right)\sum_{j,j'=1,j'\neq j}^{d}\exp(\frac{m\epsilon^2}{\sigma^2})\right) - 1\right]^{\frac{1}{2}}$$

$$= \frac{1}{2}\left[\left(\frac{d}{d^2}\exp\left(-\frac{m\epsilon^2}{\sigma^2}\right)\exp(2\frac{m\epsilon^2}{\sigma^2})\right) + \left(\frac{d(d-1)}{d^2}\exp\left(-\frac{m\epsilon^2}{\sigma^2}\right)\exp(\frac{m\epsilon^2}{\sigma^2})\right) - 1\right]^{\frac{1}{2}}$$

$$= \frac{1}{2}\left[\frac{1}{d}\exp\left(\frac{m\epsilon^2}{\sigma^2}\right) + \left(\frac{d-1}{d}\right) - 1\right]^{\frac{1}{2}}$$

$$= \frac{1}{2}\left[\frac{1}{d}\exp\left(\frac{m\epsilon^2}{\sigma^2}\right) + \left(\frac{d-1}{d}\right) - \frac{d}{d}\right]^{\frac{1}{2}}$$

$$= \frac{1}{2}\left[\frac{1}{d}\exp\left(\frac{m\epsilon^2}{\sigma^2}\right) - \frac{1}{d}\right]^{\frac{1}{2}}$$

$$= \frac{1}{2\sqrt{d}}\left[\exp\left(\frac{m\epsilon^2}{\sigma^2}\right) - 1\right]^{\frac{1}{2}}$$

$$\leq \frac{1}{2\sqrt{d}}\left[\exp\left(\frac{m\epsilon^2}{\sigma^2}\right)\right]^{\frac{1}{2}} \quad (\text{because } \sqrt{a-1} \leq \sqrt{a})$$

$$= \frac{1}{2\sqrt{d}}\left[\exp\left(\frac{\epsilon^2}{\sigma^2}\right)\right]^{m/2}$$

This gives us the final result:

$$\min_{\Psi}\max_{\substack{P_0 \in H_0 \\ P_1 \in H_1}} P_e \geq 1 - TV(P \parallel Q)$$

$$\Rightarrow \min_{\Psi}\max_{\substack{P_0 \in H_0 \\ P_1 \in H_1}} P_e \geq 1 - \frac{1}{2\sqrt{d}}\left[\exp\left(\frac{\epsilon^2}{\sigma^2}\right)\right]^{m/2}$$

$\square$

### E.3 PROOF FOR LAPLACE BOP

Here, we do the proof for a real-valued cost function, assuming that the BoP is another random variable. We consider: a Laplace distribution (for more peaked than the normal variable. We note that a similar analysis can be done for a Gamma distribution (for purely positive distributions).

**Theorem 6** (Lower bound for real-valued cost function). *The lower bound writes:*

$$\min_{\Psi}\max_{\substack{P_{\mathbf{X},\mathbf{S},Y} \in H_0 \\ Q_{\mathbf{X},\mathbf{S},Y} \in H_1}} P_e \geq 1 - \left[\frac{1}{2}\exp\left(-\frac{\sqrt{2}m\epsilon}{\sigma}\right) - \frac{1}{2}\right]$$

*where $P_{\mathbf{X},\mathbf{S},Y}$ is a distribution of data, for which the generic model $h_0$ performs better, i.e., the true $\gamma$ is such that $\gamma(h_0, h_p, \mathcal{D}) < 0$, and $Q_{\mathbf{X},\mathbf{S},Y}$ is a distribution of data points for which the personalized model performs better, i.e., the true $\gamma$ is such that $\gamma(h_0, h_p, \mathcal{D}) > 0$.*

**Corollary 4** (Maximum number of attributes (real valued cost function))**.** *If we wish to maintain a probability of error such that* $\min \max P_e \leq 1/2$ *then the number of attributes $k$ should be chosen below a value $k_{\max}$ that depends on the number of samples $N$.*

$$k_{max} \leq \frac{1}{2} - 1.4427 \log \left( \frac{0.693147\sigma}{\epsilon N} \right) \tag{27}$$

*where $\epsilon < 0$.*

We start by considering a Laplace distribution of the BoP. The proof stays the same until designing our distributions $Q$ and $P$.

**Design $Q$.** Next, we design a distribution $Q$ on this set of events that will (barely) verify $H_1$, i.e., such that the expectation of $B$ according to $Q$ will give $\gamma = 0$. We recall that $\gamma = 0$ means that the minimum benefit across groups is 0, implying that there might be some groups that have a $> 0$ benefit.

Given $p = \text{Laplace}\,(0, b) = \text{Laplace}\left(0, \frac{\sigma}{\sqrt{2}}\right)$ a centered Laplacian distribution with scale parameter $b$, we propose the following distribution for $Q$:

$$Q_j(b_j) = \prod_{k=1}^{m} p(b_j^{(k)}), \text{ for every group } j = 1....d$$

$$Q(b) = \prod_{j=1}^{d} Q_j(b_j).$$

We verify that we have designed $Q$ correctly, i.e., we verify that $Q \in H_1$. When the dataset is distributed according to $Q$, we have:

$$\begin{aligned}
\gamma &= \min_{s \in S} C_s(h_0, s) - C_s(h_p, s) \\
&= \min_{s \in S} \mathbb{E}_Q[\ell(h_0(\mathbf{X}), Y) \mid \mathbf{S} = \mathbf{s}] - \mathbb{E}_Q[\ell(h_p(\mathbf{X}), Y) \mid \mathbf{S} = \mathbf{s}] \text{ (by definition of group cost)} \\
&= \min_{s \in S} \mathbb{E}_Q[\ell(h_0(\mathbf{X}), Y) - \ell(h_p(\mathbf{X}), Y) \mid \mathbf{S} = \mathbf{s}] \text{ (by linearity of expectation)} \\
&= \min_{s \in S} \mathbb{E}_Q[B \mid \mathbf{S} = \mathbf{s}] \text{ (by definition of random variable } B) \\
&= \min_{s \in S} 0 \text{ (by definition of the probability distribution on } B) \\
&= 0.
\end{aligned}$$

Thus, we find that $\gamma = 0$ which means that $\gamma \geq 0$, i.e., $Q \in H_1$.

**Design $P$.** Next, we design $P$ as a small modification of the distribution $Q$, that will just be enough to get $P \in H_0$. We recall that $P \in H_0$ means that $\gamma \leq \epsilon$ where $\epsilon < 0$ in the flipped hypothesis test. This means that, under $H_0$, there is one group that suffers a decrease of performance of $|\epsilon|$ because of the personalized model.

Given $p = \text{Laplace}\left(0, \frac{\sigma}{\sqrt{2}}\right)$ a centered Laplacian distribution, and $p^\epsilon = \text{Laplace}\left(\epsilon, \frac{\sigma}{\sqrt{2}}\right)$ a Laplacian distribution of same variance but negative mean $\epsilon < 0$, we have:

$$P_j(b_j) = \prod_{k=1}^{m} p(b_j^{(k)}), \text{ for every group } j = 1....d,$$

$$P_j^\epsilon(b_j) = \prod_{k=1}^{m} p^\epsilon(b_j^{(k)}), \text{ for every group } j = 1....d,$$

$$P(b) = \frac{1}{d} \sum_{j=1}^{d} P_j^\epsilon(b_j) \prod_{j' \neq j} P_{j'}(b_{j'}).$$

Intuitively, this distribution represents the fact that there is one group for which the personalized model worsen performances by $|\epsilon|$. We assume that this group can be either group 1, or group 2,

etc, or group $d$, and consider these to be disjoint events: i.e., exactly only one group suffers the $|\epsilon|$ performance decrease. We take the union of these disjoint events and sum of probabilities using the Partition Theorem (Law of Total Probability) in the definition of $P$ above.

We verify that we have designed $P$ correctly, i.e., we verify that $P \in H_0$. When the dataset is distributed according to $P$, we have:

$$
\begin{aligned}
\gamma &= \min_{s \in S} C_s(h_0, s) - C_s(h_p, s) \\
&= \min_{s \in S} \mathbb{E}_P[B \mid \mathbf{S} = \mathbf{s}] \text{ (same computations as for } Q \in H_1) \\
&= \min(\epsilon, 0, ..., 0) \text{ (since exactly one group has mean } \epsilon) \\
&= \epsilon \text{ (since } \epsilon < 0).
\end{aligned}
$$

Thus, we find that $\gamma = \epsilon$ which means that $\gamma \leq 0$, i.e., $P \in H_0$.

**Compute total variation** $TV(P \parallel Q)$**.** We have verified that $Q \in H_1$ and that $P \in H_0$. We use these probability distributions to compute the lower bound to $P_e$. First, we compute their total variation:

$$
\begin{aligned}
TV(P \parallel Q) &= \frac{1}{2} \int_{b_1, ..., b_j} |P(b_1, ..., b_j) - Q(b_1, ..., b_j)| \, db_1 ... db_j \text{ (TV for probability density functions)} \\
&= \frac{1}{2} \int_{b_1, ..., b_j} \left| \frac{1}{d} \sum_{j=1}^{d} P_j^{\epsilon}(b_j) \prod_{j' \neq j} P_{j'}(b_{j'}) - \prod_{j=1}^{d} Q_j(b_j) \right| db_1 ... db_j \text{ (definition of } P, Q) \\
&= \frac{1}{2} \int_{b_1, ..., b_j} \left| \frac{1}{d} \sum_{j=1}^{d} \frac{P_j^{\epsilon}(b_j)}{P_j(b_j)} \prod_{j'=1}^{d} P_{j'}(b_{j'}) - \prod_{j=1}^{d} Q_j(b_j) \right| db_1 ... db_j \text{ (adding missing } j' = j) \\
&= \frac{1}{2} \int_{b_1, ..., b_j} \left| \frac{1}{d} \sum_{j=1}^{d} \frac{P_j^{\epsilon}(b_j)}{P_j(b_j)} \prod_{j'=1}^{d} Q_{j'}(b_{j'}) - \prod_{j=1}^{d} Q_j(b_j) \right| db_1 ... db_j \text{ (} P_j = Q_j \text{ by construction)} \\
&= \frac{1}{2} \int_{b_1, ..., b_j} \prod_{j=1}^{d} Q_j(b_j) \left| \frac{1}{d} \sum_{j=1}^{d} \frac{P_j^{\epsilon}(b_j)}{P_j(b_j)} - 1 \right| db_1 ... db_j \text{ (extracting the product)} \\
&= \frac{1}{2} \mathbb{E}_Q \left[ \left| \frac{1}{d} \sum_{j=1}^{d} \frac{P_j^{\epsilon}(b_j)}{P_j(b_j)} - 1 \right| \right] \text{ (recognizing an expectation with respect to } Q) \\
&= \frac{1}{2} \mathbb{E}_Q \left[ \left| \frac{1}{d} \sum_{j=1}^{d} \frac{\prod_{k=1}^{m} p^{\epsilon}(b_j^{(k)})}{\prod_{k=1}^{m} p(b_j^{(k)})} - 1 \right| \right] \text{ (definition of } P_j \text{ and } P_j^{(\epsilon)})
\end{aligned}
$$

**Plug in the Laplacian assumption.** Under the assumption that the random variable $B$ follows a Laplacian distribution, we continue the computations as:

$$TV(P \parallel Q) = \frac{1}{2} \mathbb{E}_Q \left[ \left| \frac{1}{d} \sum_{j=1}^{d} \frac{\prod_{k=1}^{m} \exp(-\frac{\sqrt{2}|b_j^{(k)} - \epsilon|}{\sigma})}{\prod_{k=1}^{m} \exp(-\frac{\sqrt{2}|b_j^{(k)}|}{\sigma})} - 1 \right| \right] \quad \text{(definition of } p \text{ and } p^{(\epsilon)})$$

$$= \frac{1}{2} \mathbb{E}_Q \left[ \left| \frac{1}{d} \sum_{j=1}^{d} \exp\left( -\frac{\sum_{k=1}^{m} \sqrt{2} \left( |b_j^{(k)} - \epsilon| - |b_j^{(k)}| \right)}{\sigma} \right) - 1 \right| \right] \quad \text{(property of exp)}$$

$$= \frac{1}{2} \mathbb{E}_Q \left[ \left| \frac{1}{d} \sum_{j=1}^{d} \exp\left( -\frac{\sqrt{2} \sum_{k=1}^{m} \left( |b_j^{(k)} - \epsilon| - |b_j^{(k)}| \right)}{\sigma} \right) - 1 \right| \right]$$

Since we are finding the worst case lower bound, we will find functions that upper and lower bound $|b_j^{(k)} - \epsilon| - |b_j^{(k)}|$. This function is lower bounded by $\epsilon$ and upper bounded by $-\epsilon$ since $\epsilon < 0$. To maximize $P_e$, we take the function that gives us the lower bound of $TV(P \parallel Q)$. Continuing by plugging in to get the lower bound:

$$\leq \frac{1}{2} \mathbb{E}_Q \left[ \left| \frac{1}{d} \sum_{j=1}^{d} \exp\left( -\frac{\sqrt{2} \sum_{k=1}^{m} -\epsilon}{\sigma} \right) - 1 \right| \right]$$

$$= \frac{1}{2} \mathbb{E}_Q \left[ \left| \frac{1}{d} \sum_{j=1}^{d} \exp\left( \frac{\sqrt{2} m \epsilon}{\sigma} \right) - 1 \right| \right]$$

$$= \frac{1}{2} \mathbb{E}_Q \left[ \left| \exp\left( \frac{\sqrt{2} m \epsilon}{\sigma} \right) - 1 \right| \right]$$

$$= \frac{1}{2} \left[ \exp\left( \frac{\sqrt{2} m \epsilon}{\sigma} \right) - 1 \right] \quad \text{(since all values are constant)}$$

This gives us the final result:

$$\min_{\Psi} \max_{\substack{P_0 \in H_0 \\ P_1 \in H_1}} P_e \geq 1 - TV(P \parallel Q)$$

$$\Rightarrow \min_{\Psi} \max_{\substack{P_0 \in H_0 \\ P_1 \in H_1}} P_e \geq 1 - \left[ \frac{1}{2} \exp\left( \frac{\sqrt{2} m \epsilon}{\sigma} \right) - \frac{1}{2} \right]$$

## F    COMPARISON BoP FOR PREDICTION AND BoP FOR EXPLAINABILITY PROOFS

**Proof for Theorem 3:**

*Proof.* Let $\mathbf{X} = (x_1, x_2)$ where $x_1$ and $x_2$ are independent and each follows $\text{Unif}(-\frac{1}{2}, \frac{1}{2})$. Let us define $S \in \{0, 1\}$ as $S = \mathbb{1}(X_1 + X_2 > 0)$ and $Y = S$. Then, $h_0(x) = \mathbb{1}(X_1 + X_2 > 0)$ and $h_p(x) = \mathbb{1}(S > 0)$ can both achieve perfect accuracy. Therefore, $\text{BoP}(h_0, h_p) = 0$.

For explanation, let us assume $r = 1$. Then, for model $h_0$, its important feature set $J_0$ will be either $\{X_1\}$ or $\{X_2\}$, and without loss of generality, let $J_0 = \{X_1\}$. For the personalized model, $J_p = \{S\}$.

Then, comprehensiveness of $h_0$ is

$$
\begin{aligned}
\Pr(h_0(X) \neq h_0(X_{\setminus J_0})) &= \Pr(X_1 + X_2 \leq 0 | X_2 > 0)\Pr(X_2 > 0) \\
&\quad + \Pr(X_1 + X_2 > 0 | X_2 \leq 0)\Pr(X_2 \leq 0) \qquad (28) \\
&= \Pr(X_1 + X_2 \leq 0 | X_2 > 0) \cdot \frac{1}{2} + \Pr(X_1 + X_2 > 0 | X_2 \leq 0) \cdot \frac{1}{2} \\
&= \Pr(X_1 + X_2 \leq 0 | X_2 > 0) \quad \text{(due to symmetry of the distribution)} \\
&= \int_{x_2 > 0, x_1 + x_2 \leq 0} \Pr(x_1, x_2) dx_1 dx_2 / \Pr(X_2 > 0) \\
&= 2 \cdot \int_{x_2=0}^{\frac{1}{2}} \Pr(x_2) \int_{x_1 \leq -x_2} \Pr(x_1) dx_1 dx_2 \\
&= 2 \cdot \int_{x_2=0}^{\frac{1}{2}} \Pr(x_2)(-x_2 + \frac{1}{2}) dx_2 \\
&= 2 \cdot \left[ -\frac{1}{2}x_2^2 + \frac{1}{2}x_2 \right]_0^{\frac{1}{2}} \\
&= \frac{1}{4}.
\end{aligned}
$$

For $h_p$, comprehensiveness is :

$$
\Pr(h_p(X, S) \neq h_p(X_{\setminus J_p}, S_{\setminus J_p})) = \frac{1}{2},
$$

as without $S$, $h_p$ can only make a random guess. Hence, BoP-X in terms of comprehensiveness is $\frac{1}{4}$.

For sufficiency, we can do a similar analysis:

$$
\begin{aligned}
\Pr(h_0(X) \neq h_0(X_{J_0})) &= \Pr(X_1 + X_2 \leq 0 | X_1 > 0)\Pr(X_1 > 0) \\
&\quad + \Pr(X_1 + X_2 > 0 | X_1 \leq 0)\Pr(X_1 \leq 0) \qquad (29) \\
&= \frac{1}{4}.
\end{aligned}
$$

Again, due to symmetry, equation 29 is the same as equation 28. On the other hand, the sufficiency for $h_p$ is

$$
\Pr(h_p(X, S) \neq h_p(X_{J_p}, S_{J_p})) = 0,
$$

as $J_p = \{S\}$ is sufficient to make a prediction for $h_p$. Thus, BoP-X in terms of sufficiency is also $\frac{1}{4}$.

$h_p(X)$ = random guess

$\square$

**Proof for Lemma 2:**

*Proof.* A Bayes optimal regressor using a subset of variables from indices in $J \subseteq [1, \ldots, t+k]$ would be given as:

$$
\hat{y} = h_J^*(\mathbf{x}_J, \mathbf{s}_J) = \sum_{\substack{j \in J, \\ j \leq t}} \alpha_j x_j + \sum_{\substack{j \in J, \\ j \geq t+1}} \alpha_j S_{j-t}, \qquad (30)
$$

where $h_J^*$ represents an Bayes optimal regressor for the given subset $J$, and $\mathbf{x}_J$ and $\mathbf{x}_J$ are sub-vectors of $\mathbf{x}$ and $\mathbf{s}$, using the indices in $J$. Then, the MSE of $h_J^*$ is given as:

$$
\text{MSE}(h_J^*) = \sum_{\substack{j \in \setminus J, \\ j \leq t}} \alpha_j^2 \text{Var}(X_j) + \sum_{\substack{j \in \setminus J, \\ j \geq t+1}} \alpha_j^2 \text{Var}(S_{j-t}), \qquad (31)
$$

where $\backslash J$ is a shorthand notation for $[1, \ldots t + k] \setminus J$. By combining equation 30 and equation 31, we can obtain:

$$\text{MSE}(h_0) = \sum_{j=t+1}^{t+k} \alpha_j^2 \text{Var}(S_{t+j}) + \text{Var}(\epsilon), \tag{32}$$

$$\text{MSE}(h_p) = \text{Var}(\epsilon). \tag{33}$$

We define $J_0$ and $J_p$ as a set of important features for $h_0$ and $h_p$. Note that $J_0$ and $J_p$ are the same across all samples for the additive model. Then, for regressors for sufficiency, we can write the MSE as:

$$\text{MSE}(h_{0,J}) = \sum_{\substack{j \in \backslash J_0, \\ j \leq t}} \alpha_j^2 \text{Var}(X_t) + \sum_{j=t+1}^{t+k} \alpha_j^2 \text{Var}(S_{j-t}) + \text{Var}(\epsilon) \tag{34}$$

$$\text{MSE}(h_{p,J}) = \sum_{\substack{j \in \backslash J_p, \\ j \leq t}} \alpha_j^2 \text{Var}(X_t) + \sum_{\substack{j \in \backslash J_p, \\ j \geq t+1}} \alpha_j^2 \text{Var}(S_{j-t}) + \text{Var}(\epsilon). \tag{35}$$

Similarly, for regressors for incomprehensiveness, MSE can be written as:

$$\text{MSE}(h_{0,\backslash J}) = \sum_{\substack{j \in J_0, \\ j \leq t}} \alpha_j^2 \text{Var}(X_t) + \sum_{j=t+1}^{t+k} \alpha_j^2 \text{Var}(S_{j-t}) + \text{Var}(\epsilon), \tag{36}$$

$$\text{MSE}(h_{p,\backslash J}) = \sum_{\substack{j \in J_p, \\ j \leq t}} \alpha_j^2 \text{Var}(X_t) + \sum_{\substack{j \in J_p, \\ j \geq t+1}} \alpha_j^2 \text{Var}(S_{j-t}) + \text{Var}(\epsilon). \tag{37}$$

Then, our assumption of BoP-X $= 0$ for sufficiency becomes:

$$\text{MSE}(h_0) - \text{MSE}(h_{0,J}) = \text{MSE}(h_p) - \text{MSE}(h_{p,J}). \tag{38}$$

We can expand $\text{MSE}(h_0) - \text{MSE}(h_{0,J})$ as:

$$\begin{aligned}
\text{MSE}(h_0) - \text{MSE}(h_{0,J}) &= \text{MSE}(h_0) \left( 1 - \frac{\text{MSE}(h_{0,J})}{\text{MSE}(h_0)} \right) \\
&= \text{MSE}(h_0) \left( 1 - \frac{\text{Var}(\backslash J_0) + \text{Var}(S) + \text{Var}(\epsilon)}{\text{Var}(S) + \text{Var}(\epsilon)} \right) \\
&= \text{MSE}(h_0) \frac{\text{Var}(\backslash J_0)}{\text{Var}(S) + \text{Var}(\epsilon)} \\
&= \text{MSE}(h_0) \frac{\text{Var}(J_0) + \text{Var}(\backslash J_0)}{\text{Var}(S) + \text{Var}(\epsilon)} \frac{\text{Var}(\backslash J_0)}{\text{Var}(J_0) + \text{Var}(\backslash J_0)},
\end{aligned} \tag{39}$$

where we use the shorthand notations:

$$\text{Var}(J_0) = \sum_{\substack{j \in J_0, \\ j \leq t}} \alpha_j^2 \text{Var}(X_t),$$

$$\text{Var}(\backslash J_0) = \sum_{\substack{j \in \backslash J_0, \\ j \leq t}} \alpha_j^2 \text{Var}(X_t),$$

$$\text{Var}(S) = \sum_{j=t+1}^{t+k} \alpha_j^2 \text{Var}(S_{t+j}).$$

Further, note that

$$\text{Var}(J_0) + \text{Var}(\backslash J_0) = \sum_{\substack{j \in J_0, \\ j \leq t}} \alpha_j^2 \text{Var}(X_t) + \sum_{\substack{j \in \backslash J_0, \\ j \leq t}} \alpha_j^2 \text{Var}(X_t) = \sum_{j=1}^{t} \alpha_j^2 \text{Var}(X_j) = \text{Var}(X).$$

Defining $M(h_0) \triangleq \text{MSE}(h_0)\frac{\text{Var}(X)}{\text{Var}(S)+\text{Var}(\epsilon)}$ and $r_0 \triangleq \frac{\text{Var}(J_0)}{\text{Var}(J_0)+\text{Var}(\backslash J_0)}$, we further simplify equation 39 as:

$$\text{MSE}(h_0) - \text{MSE}(h_{0,J}) = M(h_0)(1 - r_0). \tag{40}$$

Through a similar process, we can simpilfy $\text{MSE}(h_p) - \text{MSE}(h_{p,J})$ as:

$$\text{MSE}(h_p) - \text{MSE}(h_{0,p}) = M(h_p)(1 - r_p), \tag{41}$$

where $M(h_p) \triangleq \frac{\text{Var}(X)+\text{Var}(S)}{\text{Var}(\epsilon)}\text{MSE}(h_p)$ and $r_p \triangleq \frac{\text{Var}(J_p)}{\text{Var}(J_p)+\text{Var}(\backslash J_p)}$ Using equation 40 and equation 41, we arrive at:

$$M(h_0)(1 - r_0) = M(h_p)(1 - r_p). \tag{42}$$

By taking similar steps using comprehensiveness, we can derive:

$$M(h_0)r_0 = M(h_p)r_p. \tag{43}$$

By combining equation 42 and equation 43, we can conclude that:

$$\frac{r_0}{r_p} = \frac{1 - r_0}{1 - r_p} \implies r_0 = r_p.$$

Plugging this back to equation 42, we get: $M(h_0) = M(h_p)$. Now, let us assume that BoP-P $> 0$, and prove it by contradiction. Comparing equation 32 and equation 33, we can deduce that BoP-P $> 0$ means $\text{Var}(S) > 0$. Expanding $M(h_0) = M(h_p)$, we get:

$$\text{MSE}(h_0)\frac{\text{Var}(X)}{\text{Var}(S) + \text{Var}(\epsilon)} = \frac{\text{Var}(X) + \text{Var}(S)}{\text{Var}(\epsilon)}\text{MSE}(h_p),$$

$$\text{MSE}(h_p) = \frac{\text{Var}(X)}{\text{Var}(X) + \text{Var}(S)}\frac{\text{Var}(\epsilon)}{\text{Var}(S) + \text{Var}(\epsilon)}\text{MSE}(h_0),$$

$$= \frac{\text{Var}(X)}{\text{Var}(X) + \text{Var}(S)}\text{MSE}(h_p).$$

Since $\text{Var}(S) > 0$, this equality cannot hold. This concludes that BoP-P $= 0$. We can make the same claim with similar logic for a classifier where $Y$ is given as:

$$Y = \mathbb{1}(\alpha_1 X_1 + \cdots \alpha_t X_t + \alpha_{t+1}S_1 + \cdots + \alpha_{t+k}S_k + \epsilon > 0) \tag{44}$$

$\square$

## G  TRAINING DATA EXPERIMENT RESULTS

| Group | $n$ | Prediction $\hat{C}(h_0) - \hat{C}(h_p)$ | Incomprehensiveness $\hat{C}(h_0) - \hat{C}(h_p)$ | Sufficiency $\hat{C}(h_0) - \hat{C}(h_p)$ |
|---|---|---|---|---|
| Female, NW | 688 | -0.0974 | -0.1759 | -0.2718 |
| Female, W | 651 | -0.1183 | -0.2535 | -0.2197 |
| Male, NW | 657 | -0.0548 | -0.1370 | -0.1583 |
| Male, W | 654 | -0.0856 | -0.1728 | -0.1391 |
| **Total** | 2650 | -0.0891 | -0.1845 | -0.1981 |

Table 2: Evaluating A Classification Model: All metrics use 0-1 loss cost function and are found on the training dataset. The results in this table are striking, in that personalization worsens model performance across all metrics.

| Group | $n$ | Prediction | Incomprehensiveness | Sufficiency |
|---|---|---|---|---|
| | | $\hat{C}(h_0) - \hat{C}(h_p)$ | $\hat{C}(h_0) - \hat{C}(h_p)$ | $\hat{C}(h_0) - \hat{C}(h_p)$ |
| Female, NW | 688 | -0.0022 | 1.5514 | 3.8769 |
| Female, W | 651 | -0.0043 | 1.5113 | 3.2429 |
| Male, NW | 657 | -0.0032 | 1.5606 | 4.3278 |
| Male, W | 654 | -0.0143 | 1.2517 | 3.4145 |
| **Total** | 2650 | -0.0059 | 1.4699 | 3.7188 |

Table 3: Evaluating A Regression Model: All metrics use square error loss cost function and are found on the training dataset. As shown, $h_p$ assigns less accurate predictions for all groups. It decreases population prediction accuracy. For explainability, the personalized models improves incomprehensiveness for all subgroups. It leads to an overall improvement in incomprehensibility. $h_p$ improves sufficiency for all groups and overall. This example highlights that while personalization may not improve prediction accuracy, it can lead to improvements in model explainability.

## H MAX ATTRIBUTES

*Proof.* If $\min \max P_e \leq 1/2$, then:

$$1 - \frac{1}{2\sqrt{d}} \exp\left(\frac{m\epsilon^2}{2\sigma^2}\right) \leq \min \max P_2 \leq 1/2 \qquad (45)$$

Or equivalently, if $\min \max P_e \leq 1/2$, then:

$$\frac{1}{\sqrt{d}} \exp\left(\frac{m\epsilon^2}{2\sigma^2}\right) \geq 1 \qquad (46)$$

Given the number of groups $d = 2^k$, the number of samples per group $m = n/d$, the total number of samples $n = 10^4$ and the threshold of $\epsilon = 0.01$, we get:

$$\phi(k) = 1 - \frac{1}{2^{k/2+1}} \exp\left(\frac{\frac{10^4}{2^k} \times 0.0001}{2\sigma^2}\right) = 1 - \frac{1}{2^{k/2+1}} \exp\left(\frac{1}{2^{k+1}\sigma^2}\right) \qquad (47)$$

We prove that this function is an increasing function in $k$. Indeed, consider the auxiliary function $f(x) = \frac{1}{2\sqrt{x}} \exp(\frac{a}{x})$. Its derivative is $f'(x) = -\frac{\exp(\frac{a}{x})(2a+x)}{4x^{5/2}}$. For $x, a > 0$, we have: $f'(x) < 0$, i.e., $f$ is a monotonically decreasing function. Consequently, $1 - f$ is a monotonically increasing function. Thus, the function $k \to 1 - f(2^k)$ with $a = \frac{1}{2\sigma^2}$ is a monotonically increasing function of $k > 0$. □

## I MAXIMUM ATTRIBUTES (REAL-VALUED COST FUNCTION) FOR ALL PEOPLE

**Corollary 5** (Maximum attributes (real-valued cost function) for all people). *See Appendix X. Consider auditing a personalized classifier $h_p$ to verify if it provides a gain of $\epsilon = 0.01$ to each group on an auditing dataset D. Consider an auditing dataset with $\sigma = 0.1$ and $N = 8 \times 10^9$ samples, or one sample for each person on earth. If $h_p$ uses more than $k \geq 22$ binary group attributes, then for any hypothesis test there will exist a pair of probability distributions $P_{X,G,Y} \in H_0$, $Q_{X,G,Y} \in H_1$ for which the test results in a probability of error that exceeds $50\%$.*

$$k \geq 22 \implies \min_{\Psi} \max_{\substack{P_{X,G,Y} \in H_0 \\ Q_{X,G,Y} \in H_1}} P_e \geq \frac{1}{2}. \qquad (48)$$

## J EXPERIMENT PLOTS

In the following section, we show supplementary plots for the regression task on the auditing dataset. We show the distribution of the BoP across participants for all three metrics we evaluate, displaying a roughly Gaussian distribution. Additionally, we show how incomprehensiveness and sufficiency change for the number of important attributes $r$ that are kept are removed.

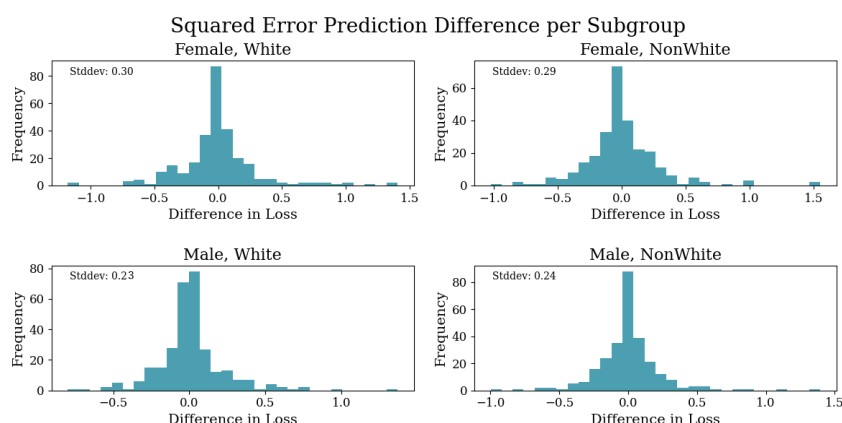

Figure 4: Individual prediction cost for all groups using the square error loss function.

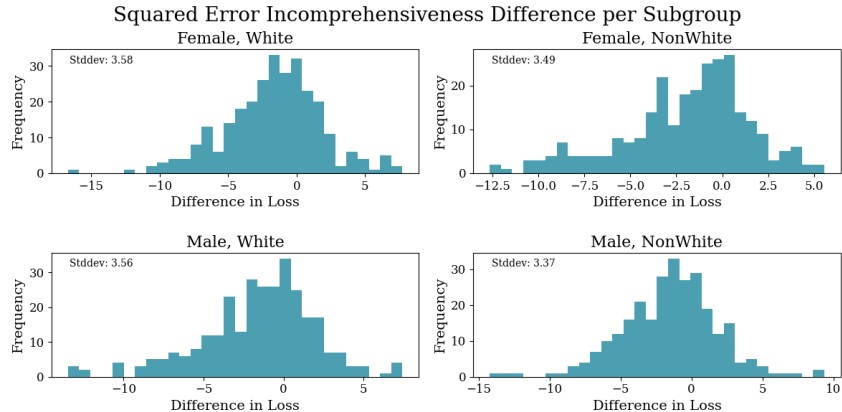

Figure 5: Individual incomprehensiveness cost for all groups using the square error loss function.

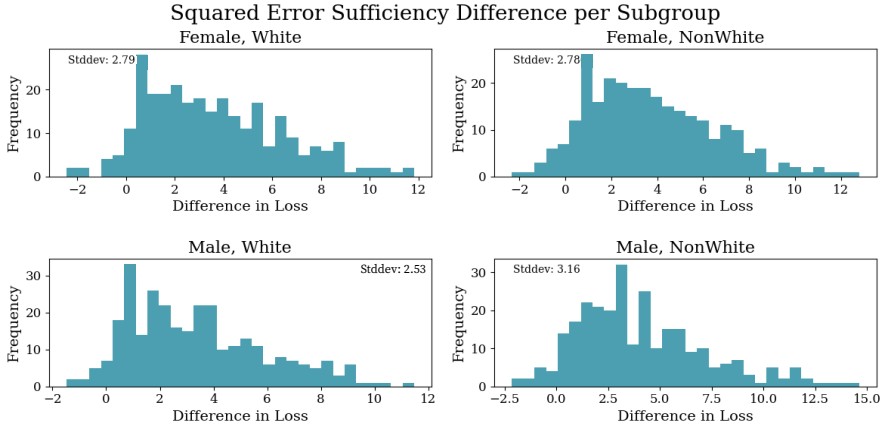

Figure 6: Individual sufficiency cost for all groups using the square error loss function.

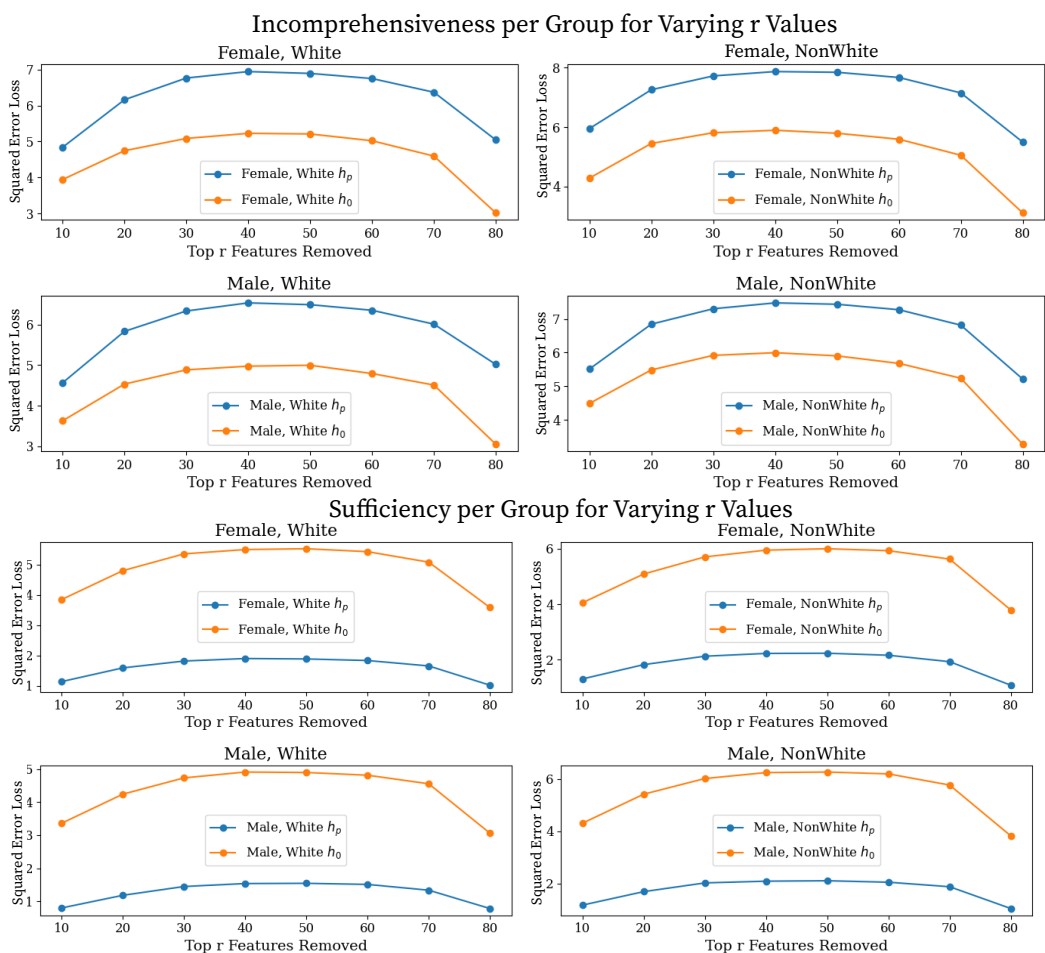

Figure 7: Values of Sufficiency and Incomprehensiveness across varying $r$ top features selected using the square error loss function. Values are found for $h_0$ and $h_p$.