# OpenReview forum: "Understanding Benefit of Personalization: Beyond Classification"
_ICLR.cc/2025/Conference — ICLR 2025 Conference Withdrawn Submission_

### Official Review · Reviewer_vrhM · 2024-11-04

**Soundness:** 2
**Presentation:** 2
**Contribution:** 2
**Rating:** 3
**Confidence:** 4

**Summary:**

The authors extend the work done by [Monteiro Paes et al. (2022)](https://proceedings.neurips.cc/paper_files/paper/2022/hash/0cfc9404f89400c5ed897035e0d3748c-Abstract-Conference.html), measuring the benefit of personalization (BoP) in regression contexts and post-hoc explanations. The empirical results highlight the potential for disparate impacts of personalization on post-hoc explanation performance across groups.

**Strengths:**

Technical contribution section is solid (note that I didn't go through them in too much detail).

**Weaknesses:**

The framing of the paper leaves a lot to be desired. Two main contributors are (1) introduction/problem statement, and (2) experiments.

(1) After reading the introduction, I, as a reader, am still not convinced on that we need the BoP metric for explanations. Rather than relying on [Balagopalan et al. (2022)](https://dl.acm.org/doi/10.1145/3531146.3533179), I would like to see the paper be more self-contained in its motivation. The fact that figure 1 is difficult to parse doesn't help either (more comments on figure 1 later).

(2) Is the experiment just meant to be a proof of concept? I would have liked to see more datasets and explainability methods to motivate BoP-X further.

As a result, it feels like the authors have added BoP-X for the sake of including more novel technical contributions.

Furthermore, the notation is a bit messy. Some of the definitions feel like they are there for the sake of having definitions (i.e. the different loss functions). I understand that the authors don't want to copy notation directly from Monteiro Paes et al. (2022), but some of it just seems redundant and unnecessary.

As for Figure 1, there is a lot going on. The figure is especially hard to understand since sufficiency and comprehensiveness is defined in Section 3 (page 5). The caption (which is already really long) needs to intuitively explain the two metrics OR put the figure after you have introduced the them.

Questions and minor points/suggestions
- Why is it a "necessity" for clinical decision making (line 82)? This is a very strong claim.
- Would be better if the paper makes it clear that we are dealing with "local" explanations and not global, model-level explanations. I know it is implied in the notation.
- Line 96 "statistical bounds": what bounds? (I know its in the abstract, but it should be here too?)
- Are sufficiency and comprehensiveness "widely-used" (line 70)? Could you point me to other works that use this metric?
- Line 135: "human-interpretable model". Why not just use the term from the paper directly: "interpretable surrogate model"?
- Line 363: Why not point to Appendix J for the reason behind the normal assumption?
- Idea: express loss/performance as empirical risk?
- Highlight statistically significant (or insignificant) results in Table 1?

**Questions:**

See above.

---

### Official Review · Reviewer_c2GE · 2024-11-06

**Soundness:** 3
**Presentation:** 4
**Contribution:** 4
**Rating:** 8
**Confidence:** 4

**Summary:**

This paper provides methods and metrics to evaluate the effectiveness of personalisation in classification and regression. In regression, the authors show the improvements to accuracy introduced by personalisation - this is contrast to the losses in accuracy when personalisation is used.  Overall this provides an understanding of the effectiveness of personalisation, where it can be useful and where it might not be appropriate to use. Their extension of this work to model explainability also gives yet another tool to build more comprehensible and hopefully more trustworthy models.

**Strengths:**

This is a clearly written paper. The claims are supported by the theoretical frameworks developed and the experimental work provides sufficient evidence of the authors claims.

**Weaknesses:**

I see the biggest contribution of this work as the explainability part -- this can help us better audit algorithms and this is always a valued contribution. On the accuracy part, several works in fairness literature have discussed the limited scope of algorithms which improve group outcomes by only looking at the model (see Selbst 2019, Fazelpour and Lipton 2020, Mitchell et. al 2021). I am fully aware the primary focus of this work is to do a technical, model study but given the application areas proposed, a section which discusses the limitations of this methodology would have greatly complemented this work.

**Questions:**

- What are the underlying assumption of the datasets that allows one to effectively use this methodology?
- What are the foreseeable risks of using this methodology in applications such as clinical settings?

---

### Official Review · Reviewer_wxCg · 2024-11-06

**Soundness:** 1
**Presentation:** 2
**Contribution:** 2
**Rating:** 3
**Confidence:** 4

**Summary:**

The paper "Understanding Benefit of Personalization: Beyond Classification" contributes to the ongoing debate regarding the potential benefits of including personalized features (e.g., gender, race) in model training. The authors extend an existing framework called Benefit of Personalization (BoP) (Monteiro Paes et al., 2022), originally defined for assessing accuracy gains from personalization in classification models, to regression models, incorporating not only accuracy gains but also explainability gains. More precisely, the authors address two main theoretical research questions:

(A) Under what conditions is general hypothesis testing for detecting personalization gains informative/useful?
(B) How do these conditions relate when assessing accuracy versus explainability gains?

To answer these questions, the authors:
1. Define a metric $\gamma$ as the minimum model accuracy (or explainability) gain across an arbitrary collection of $d$ subgroups/partitions of the dataset achieved by adding $k$ personalized binary features to the model inputs.
2. Set up a generic inference mechanism to test the null hypothesis $H_0$ that the minimum gain $\gamma$ is less than or equal to zero, against the alternative hypothesis $H_1$ that $\gamma$ is greater than an arbitrary $\epsilon > 0$.
3. Define the sum of Type I and Type II errors for the previous test, denoted as $P_e$.
4. Derive a lower bound for the error probability $P_e$ as a function of $d$, $k$, and parameters of the distributions assumed for $\gamma$.
5. Compare the error $P_e$ lower bound to $1/2$, a minimum acceptable value for the test to be informative.
6. Analyze the relationship between $P_e$ lower bounds for explainability and accuracy gains.
7. Illustrate their methodology in models designed to predict future academic performance using the High School Longitudinal Study dataset.

**Strengths:**

Extending the BoP framework beyond classification models is a novel contribution. However, perhaps the biggest contribution is the inclusion of explainability metrics in BoP, in addition to accuracy metrics. This is important to the broader "Responsible Machine Learning" debate regarding whether or not sensitive attributes should be allowed by law in model training. As shown by the authors, it is possible to have zero accuracy gains but positive explainability gains, and providing a framework to identify scenarios when inference on BoP cannot be trusted is a step in this direction.

The appendix is well-organized, with explanations of their proofs.

**Weaknesses:**

- The paper could benefit from being more upfront from the outset about what specific question it is trying to answer: (A) under what conditions is general hypothesis testing for the detection of personalization gains informative/useful? (B) How do these conditions relate when assessing accuracy versus explainability gains?
- Related to the previous suggestion, be precise about what kinds of bounds you are deriving. I read the Abstract, Introduction (including the specific contributions), and continued to section 5, and I still had no idea what precisely the bounds being derived are and how they can be useful. It often seems like the bounds are on the BoP, but they are on the $P_e$. Monteiro Paes et al. (2022) derive bounds for their BoP, but not this paper.
- In line with the previous comments on making your contribution clearer and more precise from the outset: substantial work needs to be done to motivate and explain how to use the lower bounds on $P_e$. I understand that it allows one to identify situations when a hypothesis test on BoP isn’t reliable, however, several points need to be made explicit, motivated, and potentially worked to make the bound really useful in practice:

  (i) The bounds rely on stringent and often unrealistic assumptions, such as for datasets where the size $m$ of the subgroups is the same across groups (i.e., $m = n/d$), $d = 2^k$ subgroups, $k$ are binary attributes (can they be categorical?). Even in your application, these restrictions do not hold, making the $P_e$ bounds invalid for most cases? Is there a way to relax these assumptions?

  (ii) There is no mechanism to test the $H_0$ vs $H_1$ that you propose, i.e., there is no test statistic and an associated distribution for this test. How can the lower bounds on the errors of this test $P_e$ be useful if there isn’t a test? In other words, what can we learn about the world when we know the lower bound, but have no test?

  (iii) $P_e$, the central object in the paper, is often referred to as the error probability and bounds are being computed on this “probability.” However, $P_e$ is the sum of Type I and Type II errors, potentially adding up to more than 1, i.e., not being a probability per se. The paper you cite (Monteiro Paes et al., 2022) defines $P_e$ as the average of these two error types. If you choose to define it as the average of these two errors, you still need to motivate what it means to find a lower bound for that average, knowing that these errors are conditioned on different states of the world.

  (iv) Related to the subpoint above, the lower bounds for $P_e$ are compared to a coin flip probability of $1/2$, i.e., uniform binary Bernoulli, when supposedly the test procedure isn’t informative/useful. This needs to be explained why. $1/2$ is usually the random benchmark for predicting a binary random variable, but in the test procedure, you can commit Type I error, Type II error, or none of these errors, making the test procedure no longer a binary procedure. A more detailed explanation can be helpful.

  (v) Choice of distribution and parameters: It is mostly harmless to derive the lower bound for $P_e$ for classification models, as done by Monteiro Paes et al. (2022), which models the distribution of $\gamma$ nonparametrically as a categorical distribution. However, when deriving the bounds for regression models in your paper, assuming that $\gamma \sim N(0, \sigma^2)$ does not come without a cost and needs to be highly motivated. With the normal assumption, the bounds end up having too many degrees of freedom ($k$, $\epsilon$, and $\sigma$). How to choose them simultaneously? Particularly, how does one choose $\sigma$? For instance, you have different $\sigma$s in Figure 3. You could employ different distributions and different parameterizations to show that results aren’t sensitive to this choice. Also, $\gamma$ is highly discontinuous, as it is a minimum BoP over discrete subgroups, so assuming it is normally distributed isn’t realistic?

- A sizable amount of the paper (and the appendix) is virtually re-stating and re-deriving what was done in Monteiro Paes et al., 2022, a paper introducing BoP, published 2 years ago with 8 citations. I wonder if you can just cite this previous paper for the main definitions and derivations and shift your focus more to the explainability versus accuracy gains, which seems to be one of your contributions, making your paper about what you bring new to the table.
- Two arbitrary measures for explainability are selected, namely Faithfulness and Sufficiency, out of an infinitude of measures. Also, these measures have a great overlap/correlation (as also shown in Figure 7 in the appendix, they are highly correlated). It would be great to have some motivation and justification about why these measures were chosen.

**Questions:**

Specific questions/suggestions (please feel free to ignore these if not useful):
- The title "Understanding Benefit of Personalization: Beyond Classification" reads a bit awkward. I know you meant understanding the BoP, but I would suggest adjusting it to something like "Understanding the Benefits of Personalization: Beyond Classification."
- Typo in the Appendix, row 984: Is the Cauchy-Schwarz inequality missing a squared term?
- Equation 3 defines cost as a function of $h$, $x$, $y$, but not $s$. However, $s$ is inside the personalized model. One solution is to define that cost as a function of $h$? $h_0$ does not contain $s$, and $h_P$ does.
- Typo on row 210: "We denote the the resulting..."
- Graphs as in Figure 3 label the y-axis as $P_e$ and have the title "Classification Probability Error." Are you computing $P_e$ or the lower bound of it?
- Definitions 3 and 4 use the same notation for the cost function, which can be confusing. Perhaps use a different notation or subscript?
- Be careful with using linearly additive models to model probabilities (as in Lemma 2), as they are often not bounded between 0 and 1.
- Suggested title for Section 2: "Related Work" as opposed to "Related Works."

---

### Official Review · Reviewer_aExi · 2024-11-11

**Soundness:** 2
**Presentation:** 2
**Contribution:** 1
**Rating:** 1
**Confidence:** 3

**Summary:**

This paper contributes to the evaluation of the _Benefit of Personalization_ (BoP) of machine learning models. In particular, it extends BoP, which has classically been defined on classification tasks, to regression tasks, it extends the definition of BoP the effect of personalization on model interpretability, and it proposes a statistical framework for assessing the significance of the effect of protected attributes.

**Strengths:**

- Definitions of terms such as "Incomprehensiveness," "sufficiency," "population BoP," and "Groupwise BoP" were clearly stated throughout the paper
- The introduction section is very clear
- Derives novel theoretical bounds on the probability of error for whether personalization improves results for at least one group, and shows how it relies on the number of features. That said, I found it hard to conclude how one would practically use these bounds, and whether they are in fact robust, given the results in section 6 ("Applying the Framework") which were only evaluated on one dataset and seemed suspect (see weaknesses).

**Weaknesses:**

- I found the original/creative contributions of this paper to be minimal. The extensions of BoP to regression and explainability are fairly straightforward, and are not substantive enough contributions for the standards of ICLR.
- The paper is written in a highly technical format, and especially for a sensitive subject like how to evaluate the benefit of including sensitive attributes, the writing should be improved to make the results more accessible and interpretable to an ML practitioner. After reading the paper, I struggle to summarize the practical takeaway that I would use if I were building a model and considering using sensitive attributes as features. This is especially true in the section 5. Section 6 should have helped in this regard, but I didn't find that the authors clearly summarized the take-home messages in this section. Overall, I think the writing should be improved to make the paper more accessible and to help readers digest and be convinced of the main messages of the paper, especially in sections 5 and 6.
- In section 6, with a real world dataset, does it make sense that personalization worsens certain metrics in the classification setting but improves them in the regression setting? I have not personally been convinced enough by the previous sections to trust these results.

**Questions:**

How do you make sense of the fact that in Section 6, personalization worsens certain metrics in the classification setting but improves them in the regression setting?

---

### Note · Authors · 2024-11-20

**Comment:**

Thank you so much for your time and valuable feedback on our paper. We greatly appreciate your thorough review and the thoughtful comments provided.
After careful consideration, we have decided that the paper requires significant revisions to better highlight its contributions and align with its intended goals. Consequently, we will be reorganizing the paper and withdrawing it at this time.
We deeply value the insights you have shared and hope to incorporate them into a stronger version of the paper in the future. Thank you again!

**Withdrawal Confirmation:**

I have read and agree with the venue's withdrawal policy on behalf of myself and my co-authors.